# Technical note: c-u-curve: A method to analyse, classify and compare dynamical systems by uncertainty and complexity

Uwe Ehret[1], Pankaj Dey[2,3]

[1]Institute of Water Resources and River Basin Management, Karlsruhe Institute of Technology (KIT), Karlsruhe, Germany
[2]Department of Civil Engineering, National Institute of Technology, Sikkim, Ravangla, India
[3]Interdisciplinary Centre for Water Research, Indian Institute of Science, Bangalore, India

*Correspondence to*: Uwe Ehret (uwe.ehret@kit.edu)

**Abstract.** We propose and provide a proof-of-concept of a method to analyse, classify and compare dynamical systems of arbitrary dimension by the two key features uncertainty and complexity. It starts by subdividing the system's time-trajectory
into a number of time slices. For all values in a time slice, the Shannon information entropy is calculated, measuring within-slice variability. System *uncertainty* is then expressed by the mean entropy of all time slices. We define system *complexity* as "uncertainty about uncertainty", and express it by the entropy of the entropies of all time slices. Calculating and plotting uncertainty $u$ and complexity $c$ for many different numbers of time slices yields the *c-u-curve*. Systems can be analysed, compared and classified by the c-u-curve in terms of i) its overall shape, ii) mean and maximum uncertainty, iii) mean and
maximum complexity, and iv) its characteristic time scale expressed by the width of the time slice for which maximum complexity occurs. We demonstrate the method at the example of both synthetic and real-world time series (constant, random noise, Lorenz attractor, precipitation and streamflow) and show that the shape and properties of the respective c-u-curves clearly reflect the particular characteristics of each time series. For the hydrological time series we also show that the c-u-curve characteristics are in accordance with hydrological system understanding. We conclude that the c-u-curve method
can be used to analyse, classify and compare dynamical systems. In particular, it can be used to classify hydrological systems into similar groups, a precondition for regionalization, and it can be used as a diagnostic measure which can be used as an objective function in hydrological model calibration. Distinctive features of the method are i) that it is based on unit-free probabilities, thus permitting application to any kind of data, ii) that it is bounded, iii) that it naturally expands from single-to multivariate systems, and iv) that it is applicable to both deterministic and probabilistic value representations, permitting
e.g. application to ensemble model predictions.

**Keywords**

Complexity, uncertainty, entropy, dynamical system analysis, catchment classification, catchment similarity.

## 1 Introduction

In the earth sciences, many systems of interest are dynamical, i.e. their states are ordered by time and evolve as a function of time. The theory of dynamical systems (Forrester, 1968; Strogatz, 1994) therefore has proven useful across a wide range of earth science systems and problems such as weather prediction (Lorenz, 1969), ecology (Hastings et al., 1993; Bossel, 1986), hydrology (Koutsoyiannis, 2006), geomorphology (Phillips, 2006) and coupled human-ecological systems (Bossel, 2007).

Key characteristics of dynamical systems include their mean states (e.g. climatic mean values in the atmospheric sciences),
their variability (e.g. annual minimum and maximum streamflow in hydrology) and their complexity (e.g. population dynamics in ecological predator-prey cycles). Interestingly, despite its importance and widespread use there is to date no single agreed-upon definition and interpretation of complexity, and no agreed-upon base set of features characterizing a complex system. Gell-Mann (1995), Lloyd (2001), Prokopenko et al. (2009) and Ladyman et al. (2013) provide interesting overviews on the topic. Gell-Mann (1995) points out that while measures of complexity for entities in the real world are to

some degree always context-dependent, they have in common that "… complexity can be high only in a region intermediate between total order and complete disorder.". Lloyd (2001) provides a short yet comprehensive list of complexity measures categorized by difficulty of description, difficulty of creation, and degree of organization. Prokopenko et al. (2009) discuss the connection of complexity, self-organization, emergence and adaptation, and suggest an information-theoretic framework to promote inter- and transdisciplinary communication on these topics. Ladyman et al. (2013) review various approaches to

define complex systems, distil a set of core features common to all definitions (nonlinearity, feedback, emergence, hierarchy, numerosity, among others) and provide a large collection and a taxonomy for measures of complexity.

Characterizing dynamical systems by few and meaningful statistics representing the above-mentioned key features is important for several reasons: System classification, intercomparison and similarity analysis is a precondition for the transfer of knowledge from well-known to poorly-known systems or situations (see e.g. Wagener et al., 2007; Sawicz et al., 2011;

and Seibert et al., 2017 for applications in hydrology). Further, dynamical system analysis helps detecting and quantifying nonstationarity, a key aspect in the context of global change (Ehret et al., 2014), and it is important for evaluating the realism of dynamical system models and for guiding their targeted improvement (Moriasi et al., 2007; Yapo et al., 1998).

In this paper, we address the task of parsimonious yet comprehensive characterization of dynamical systems by proposing a method based on concepts of information theory. It comprises both variability and complexity, and adopts the view that the

overall variability (or uncertainty) of a time series is the mean of its variabilities in sub periods, and that the complexity of a time series is the overall variability of these variabilities. We use examples from hydrology, as due to the multitude of subsystems and processes involved, most hydrological systems classify as variable and complex systems (Dooge, 1986). Hydrological systems and models thereof have been analysed in terms of predictive, model structural and model parameter uncertainty by Vrugt et al. (2003), Liu and Gupta (2007) and Vrugt et al. (2009), among others. Hydrological systems have

been classified in terms of their complexity by Jenerette et al. (2012), Jovanovic et al. (2017), Ossola et al. (2015), Bras (2015), Engelhardt et al. (2009), Pande and Moayeri (2018), Sivakumar and Singh (2012), Sivakumar et al. (2007) and Omabdi et al. (2021) among others. Following early attempts by Jakeman and Hornberger (1993), Pande and Moayeri (2018) have investigated how the relation between the information content and complexity of hydrological systems can guide the selection of adequate models thereof, and vice versa.

In particular, concepts from information theory have been applied for hydrological system analysis and classification by Pachepsky et al. (2006), Hauhs and Lange (2008), Zhou et al., (2012), Castillo et al., (2015), and recently by Dey and Mujumdar (2021). Information-based approaches rely on log-transformed probability distributions of the quantities of interest, and are thus independent of the units of the data. Compared to methods relying directly on the data values, this poses an advantage in terms of generality and comparability across disciplines. Being rooted in information theory, the

method we propose in this paper makes use of this advantage. The same applies to the methods of multiscale entropy (MSE) proposed by Costa et al. (2002) in the context of physiologic time series, and the method suggested by López-Ruiz et al. (1995) for physical systems. Both share similarities with the c-u-curve method, but also differ in some important aspects, which will be discussed in Sect. 2.3, after the c-u-curve method has been introduced in Sect. 2.1. The MSE method has been applied to a wide range of complex systems, such as biological signals (Costa et al. 2005), ball bearing fault measurements

(Wu et al., 2013), seismic (Guzmán-Vargas et al., 2008) and hydro-meteorological time series. For the latter, Li and Zhang (2008) analysed long time series of Mississippi river flow data, Chou (2011) used MSE in combination with wavelet transformation to analyse properties of station-based rainfall time series. Brunsell (2010) also applied entropy measures on various temporal scales to assess spatial-temporal variability of daily precipitation, similar to the MSE method, but refers to it as "a multiscale information theory approach".

The remainder of the text is organized as follows: In Sect. 2, we present all steps of the method, describe its properties and compare it to existing methods. In Sect. 3, we apply the method to both synthetic time series and observed hydrological data

to demonstrate uses and interpretations of the c-u-curve method. We summarize the method, discuss its limitations and draw conclusions in the final Sect. 4.

## 2 Method

Please note that in what follows, for clarity we introduce the method at the example of univariate time series with deterministic values, and we calculate discrete entropy based on a uniform binning approach.

### 2.1 Method description

The mathematical variable names used in this section and throughout the paper were chosen with the goal of straightforward interpretation. The names were constructed by combination of the following base "alphabet" : $n$ is for "number", $v$ is for

"value", $b$ is for "bin", $s$ is for "(time) slice", $e$ is for "entropy", $t$ is for "time", $w$ is for "width". For example, variable $nvb$ is formed by a combination of three symbols and represents the "number of value bins". To avoid confusion of combined variable names with multiplication (e.g. $nvb$ could be falsely interpreted as the product of variables $n$, $v$, and $b$), we explicitly indicate each multiplication with the "·" symbol.

Applying the method to a given time series with overall $nt$ time steps consists of a number of steps and related choices: At

first, for each variate involved a suitable discretization (binning) scheme is chosen. The bins must cover the entire value range, and their total number $nvb$ can be chosen according to a user's demands on data-resolution. Next, the time series is divided into a number of $ns$ time slices. The slices must be mutually exclusive and together must cover the time series. The slices are preferably, but not necessarily, of uniform width. Next, separately for each slice, a discrete probability distribution (histogram) is calculated using the data in the slice and the chosen binning scheme. From the so-obtained histogram, the

Shannon information entropy $H$ (Shannon, 1948) is calculated following Eq. (1),

$$H(X) = - \sum_{vb=1}^{nvb} p(x_{vb}) \cdot \log_2(p(x_{vb})) \tag{1}$$

where $X$ is all sample data within the slice, $p(x_{vb})$ refers to the probability of variate value $x$ falling into bin $vb$, and $nvb$ is the total number of value bins. Entropy measures data variability or uncertainty in bit, with the intuitive interpretation as "the minimum number of binary (Yes/No) questions needed to be asked to correctly guess values drawn from a known distribution". Cover and Thomas (2006) provide an excellent introduction to information theory, applications to hydrology

and hydrometeorology are e.g. presented in Singh (2013) and Neuper and Ehret (2019). Neuper and Ehret (2019) also describe the relation of entropy and variance: "Like the variance of a distribution, entropy is a measure of spread, but there are some important differences: while variance takes the values of the data into account and is expressed in (squared) units of the underlying data, entropy takes the probabilities of the data into account and is measured in bit. Variance is influenced by the relative position of the data on the measure scale and dominated by values far from the mean; entropy is influenced by

the distribution of probability mass and is dominated by large probabilities. Some welcome properties of entropy are that it is applicable to data that cannot be placed on a measure scale (categorical data), and that it allows comparison of distributions from different data due to its generalized expression in bit.

As entropy values may differ between slices, an overall uncertainty estimate for all slices is calculated as the expected value of all slice entropies. For equal-width slices, this is mean entropy according to Eq. (2),

$$Uncertainty = E\big(H(X)\big) = \overline{H(X)} = \frac{1}{ns} \cdot \sum_{s=1}^{ns} H_s(X) \tag{2}$$

where $s$ refers to a particular slice of all $ns$ time slices. The so-defined *uncertainty* measures average within-slice variability of the data, i.e. uncertainty of the time series as seen through the lens of the chosen time slicing scheme.

Next, we consider variability of entropy across all slices, and as before we measure variability by entropy. In order to calculate this higher-order "entropy of entropies", a suitable binning scheme for entropy values must be chosen, which can be based on the same criteria as outlined above. It is then used to calculate a histogram of the $ns$ entropy values. We thus define *complexity* as the entropy of entropy values, which is calculated following Eq. (3),

$$Complexity = H\big(H(X)\big) = -\sum_{eb=1}^{neb} p(H_{eb}) \cdot \log_2(p(H_{eb})) \tag{3}$$

where $neb$ denotes the total number of entropy bins, $eb$ a particular entropy bin, and $p(H_{eb})$ the probability of a time slice entropy $H_s$ falling into bin $eb$. Complexity measures how uncertain we are about the uncertainty in a particular time slice, when all we know is the distribution of uncertainties (entropies) across all time slices in the time series. The question may arise why complexity is calculated as the entropy rather than the variance (= 2$^{nd}$ moment) of entropies, which would seem a logical extension of uncertainty being calculated as the mean (=1$^{st}$ moment) of entropies. There are three reasons for this choice, namely consistency, interpretability, and robustness. "Consistency" refers to the idea that when expressing the variability of the distribution within a time slice by entropy, we think that it is a natural choice to express variability of the variabilities also by entropy. Thus, variability is always expressed in the same unit of bit, which increases comparability among the values and upper bounds of uncertainty and complexity. "Interpretability" refers to the fact that entropy has the intuitive interpretation of "number of binary Yes/No questions to ask to move from a prior to a posterior state of knowledge", while variance lacks this straightforward interpretation. "Robustness" refers to the previously discussed property of variance being more sensitive to outliers in the data than entropy. While for extreme-value statistics with a focus on the tails of a distribution, variance is a good choice, we think that for a characterization of the overall variability of a data set, entropy is more appropriate.

The entire procedure of calculating uncertainty and complexity is repeated for many different choices of $ns$ (time slicing schemes). For each choice of $ns$, for equal-width slices the width of a time slice is $sw = nt/ns$. In principle, $ns$ can be chosen to take any value in the range $[1,nt]$. For $ns = 1$, the entire time series is contained in a single slice of width $sw = nt$. For $ns = nt$, each time slice contains only a single time step. However, it is recommended to choose $ns$ - and with it $sw$ - from a smaller range: If we require that for a robust estimation of a time slice histogram, each of its $nvb$ bins should on average be populated by a minimum number of $m$ values, then the width $sw$ of a time slice (i.e. the number of values within) must at least be $nvb \cdot m$ (see Eq. 4). This means that for robust estimates of *uncertainty*, the time series should be split into only few but wide time slices. For robust estimates of *complexity*, however, it is the opposite: The histogram of uncertainty values is populated by a total of $ns$ values (the entropies of all time slices). If for the sake of a robust estimation we again require that each of the histogram bins should be populated by at least $m$ values, then at least $neb \cdot m$ time slice entropy values are needed. This means that the time series should be split into many - and hence narrow - time slices. These two antagonistic constraints lead to an upper and lower limit for the choice of $sw$, which is formalized in Eq. 4: For a (subjective) user's choice of $m$, Eq. 4 yields the range of time slice widths $sw$ satisfying the "$m$-criterion" for both the uncertainty and complexity histogram as a function of time series length $nt$ and the number of bins for both uncertainty ($nvb$) and complexity ($neb$).

$$\frac{nt}{neb \cdot m} \geq sw \geq nvb \cdot m \tag{4}$$

For example, for a time series with $nt = 30000$ time steps, and choices of $m = 3$, and $nvb = neb = 10$ (all histograms resolved by ten bins), the range of suitable time slice widths is $[30,1000]$. It should be noted that Eq. 4, through the choice of $sw$, provides one possible guideline for robust histogram estimation, but a user can also resort to other binning guidelines, such as the methods suggested by Sturges (1926), Scott (1979), Freedman and Diaconis (1981), Pechlivanidis et al. (2016) or Knuth (2019). Throughout all time slicing schemes, the number of value and entropy bins must be kept constant to assure comparability. Together, the set of all time slicing schemes produces a set of complexity-uncertainty value pairs. Plotting

them with uncertainty values on the x-axis and complexity values on the y-axis is what we call the *complexity-uncertainty-curve*, or short *c-u-curve*. It summarizes several interesting properties of the time series under consideration, which will be discussed in Sect. 3.

## 2.2 Properties

In this section, we briefly summarize some general properties of the c-u-curve and discuss its limitations and possible generalizations.

*Axes units.* For the c-u-curve, both the x-axis (showing uncertainty) and the y-axis (showing complexity) are in units of bit (see Eqs. 2 and 3), i.e. they are independent of the units of the data. This facilitates intercomparison of different systems, and application to multivariate systems where variates are in different units.

*Existence of lower and upper bounds for uncertainty*. The lower bound for uncertainty is always zero, which is reached if for all time slices, all values within a time slice fall into the same value bin. The upper bound is dependent on the choice of $nvb$ (the number of bins resolving the value range). Its value, $log_2(nvb)$, is the entropy of a uniform (=maximum entropy) distribution. It is reached when the data within each time slice are uniformly distributed across all value bins. In a plot of the c-u-curve, the upper uncertainty bound appears as a vertical line.

*Existence of lower and upper bounds for complexity*. Same as for uncertainty, the lower bound of complexity is always zero. It is reached if the entropy values calculated for all time slices all fall into the same entropy bin. The upper bound is dependent on the choice of $neb$ (the number of bins resolving the entropy range). Similar to uncertainty, its value, $log_2(neb)$, is the entropy of a uniform distribution and reached when the entropies of all time slices are uniformly distributed across all entropy bins. In a plot of the c-u-curve, this global upper complexity bound appears as a horizontal line.

However, there exists another, more strict upper bound, where maximum reachable complexity is a function of uncertainty: Consider the distribution of entropy values of all time slices. Its mean value is represented by uncertainty (see Eq. 2). It poses a constraint on how the entropy values can be distributed over the entropy bins, and hence the maximum entropy this distribution can reach. For example, if the mean entropy lies within the lowest entropy bin, all entropy values necessarily also have to be placed in that bin, which corresponds to a Dirac distribution, which has an entropy of zero. Zero uncertainty

therefore necessarily implies zero complexity. The same applies if mean entropy lies in the maximum entropy bin: In that case, all entropy values necessarily have to lie in that bin, too, which again corresponds to a Dirac distribution with zero complexity. More general, the reachable upper bound for complexity is determined by solving the task to find, for a discrete (binned) probability distribution with a finite number of distinguishable states and known mean (here: uncertainty) from all possible distributions the one which maximizes entropy (here: complexity). The solution for this task has been provided by

Conrad (2022), and is summarized in Appendix B. In a plot of the c-u-curve, this upper bound for complexity appears as an arch, starting at the origin, and terminating at the upper uncertainty bound with zero complexity.

*Invariance under normalization*. The shape and values of the c-curve remain invariant under prior normalization of the data if the binning scheme is also transformed. Normalization can therefore be applied for convenience to use the same binning scheme for all time series. Likewise, for better comparability among time series of different length, normalization of the time

domain is also possible. As a consequence, the time slice widths *sw* will be expressed in units of "length relative to the length of the time series" rather than in the original time units. However, this potentially comes at the cost of losing interpretability, e.g. to detect the effect of diurnal or seasonal cycles in the c-u-curve.

*Influence of the chosen binning scheme*. The values of the bounds, and all uncertainty and complexity values of the curve depend on the chosen binning for the values and the entropies. For direct comparison of c-u-curves, the binnings should

therefore agree. If this is for some reason not feasible, comparability can be established by normalizing values to a [0,1] range. This can be achieved by dividing values of the c-u-curve with the values of the global upper bounds for uncertainty and complexity.

*No guarantee for continuity*. For better visibility, we connected the c-u points calculated for different time slice widths *sw* in Fig. 2 and Fig. 3 by a line. However, there is no theoretical argument guaranteeing continuity of the c-u-curve, and the lines should not be interpreted in this manner. Nevertheless, test runs with many different data sets and many time slice widths suggest that the c-u-curve generally is smooth.

*Influence of time slice positioning*. For short time series with highly variable data, different splits of the time series into time slices might return quite different results. In other words, the default splitting scheme starting at the first time step (e.g. "1-2-3", "4-5-6", etc. for time slices of width $sw = 3$) might not be representative for all other possible splitting schemes (e.g. "2-3-4", "5-6-7", etc.). To investigate the sensitivity of the c-u-curve results to time slice positioning, we repeated all applications as discussed in Sect. 3 with a moving-window approach, applying all possible splitting schemes, and analysed the variability of the results (not shown). For all applications, the results were almost indistinguishable from each other, the overall sensitivity to the splitting scheme therefore seems small. Nevertheless, in the c-u-curve code (Ehret, 2022) published together with this paper, the user can choose between the default splitting scheme and a moving-window approach, where all possible splitting schemes are applied and the results are averaged.

*Influence of errors and trends in the data*. Without formal proofs, we briefly discuss here the effect of errors or trends in the data on the values and shape of the c-u-curve. In the case of *random errors* coming from a particular distribution (e.g. measurement error), uncertainty about the true entropy of a time slice will be equal to the entropy of the error distribution, and, as information from independent sources is additive, the total entropy of a time slice will be the sum of the within-slice entropy without the error plus the entropy of the error distribution. As the additional entropy by the error is the same for all time slices, mean entropy of all time slices (uncertainty) will also increase by the entropy of the error, but the distribution of entropies will remain its shape, as a consequence the entropy of that distribution (complexity) will remain unchanged. Random error therefore will shift the c-u-curve to the right. A *bias* in the data will shift the distribution of the values in a time slice, but its shape will remain unchanged, and so will its entropy. As this applies to all time slices, the c-u-curve will remain unchanged. *Trends* in the data will increase the variability within all time slices in the same manner, such that uncertainty increases, but complexity remains unchanged. *Breakpoint*s in the data, where one (or no) trend is replaced by another, will increase the variability of the time slice entropies, and hence complexity.

*Generalizations and limitations*. We introduced the c-u-curve method by an univariate and deterministic example. However, the method is also applicable to multivariate and/or probabilistic data. When moving from univariate to multivariate data, the entropy within a time slice simply changes from uni- to multivariate entropy. When moving from deterministic to probabilistic variables, for each time step in a time slice, a value distribution rather than a crisp value will be used to populate the distribution of all values in the time slice, but the result will still be a single distribution with a single entropy value, which can be plotted as before in the c-u-curve. In Ehret (2022), we provide multivariate and probabilistic application examples, and the related generalized code. Also, in the method description in Sect. 2.1, we calculated discrete entropy based on a uniform binning approach. We did so as it has some useful properties (ease of interpretation is one of them) compared to calculating continuous entropy. Nevertheless, the method can also be used with non-uniform binning or continuous representations of data-distributions, as long as entropy can be calculated from the data distribution. For a detailed discussion of discrete vs. continuous entropy, see Azmi et al. (2021) and references therein. Please also note that strictly speaking, the c-u-curve method does not measure the uncertainty and complexity of an entire dynamical system, but only of its signals (time series) which are available for analysis. For cases where the signals do not completely cover the system's state space, we should therefore refer to the results as "signal uncertainty" and "signal complexity". As throughout the literature on dynamical system analysis, this distinction is usually not made, we also stick to the term "system" throughout the paper.

## 2.3 Comparison to existing methods

Two methods similar to the c-u-curve have been proposed in the literature, which we will in the following briefly explain and discuss. The first, $C_{LMC}$, was proposed by López-Ruiz (1995), the second, multiscale entropy (MSE) by Costa et al. (2002). $C_{LMC}$ is a statistical measure of complexity for physical systems. It is calculated as the product of the system's information content, which is measured by the (normalized) Shannon entropy of the probability distribution of all of its accessible discrete states, and disequilibrium, which is measured by the sum - taken over all accessible discrete states - of squared differences between the system's probability distribution and a corresponding uniform (=maximum Entropy) distribution. For example, a crystal is has high disequilibrium but low information content, and an ideal gas has low disequilibrium but high information content, but for both the product $C_{LMC}$ is small, indicating low complexity. Plotting a system's $C_{LMC}$ over its information content (see Fig. 2 in López-Ruiz, 1995) looks similar to the c-u-curve, including the limit behaviour (complexity approaches zero for systems with very high and very low entropy) and the existence of an upper bound for complexity as a function of entropy. Feldman and Crutchfield (1998) later proposed replacing the somewhat arbitrary measure of disequilibrium in López-Ruiz (1995) by Kullback-Leibler divergence, but the essential differences of $C_{LMC}$ and the c-u-curve methods remain: Firstly, the former defines complexity as the product of two separate system characteristics, of which one is the departure from a benchmark system, the latter derives both characteristics from the system alone. Secondly, the former does not take the order of the data into account, while the latter explicitly does when calculating entropy for data within temporally neighbouring data within time slices.

The MSE method calculates entropy of a time series for various coarse-grained (=time-averaged) versions thereof, and then plots entropy over the size of the averaging window (referred to a scale factor $\tau$ in Costa et al, 2002). MSE shares with the c-u-curve the idea that from the joint display and comparison of various entropy values of a time series much can be learned about the underlying dynamical system, it is also similar in that the temporal order of the data is explicitly taken into account. The main difference is that in MSE, data in a time window are averaged, i.e. the within-window variability of the data is essentially removed, while in the c-u-curve entropy calculations are always done on the original data. The second difference is that MSE does not provide an objective measure of system complexity, rather this is visually inferred from the plot: Complex systems are those revealing high entropy values across a wide range of scale factors. Obviously, the MSE and the c-u-curve approach can be joined by repeating c-u-curve calculations for various coarse-grained versions of a time series, which seems like a very promising idea for future work.

## 3 Application to synthetic and real-world time series

### 3.1 Time series description

We discuss the properties of the c-u-curve at the example of six time series as shown in Fig. 1a-f. Time series a-c are synthetic time series: a straight line, random uniform noise, and the famous Lorenz attractor (Lorenz, 1963). We chose them for their simple, exemplary and well-known behaviour. The straight line (Fig 1a) contains no variability whatsoever and should therefore show both very little uncertainty and complexity. The random noise (Fig. 1b) contains very high, but constant variability and should therefore show high uncertainty and low complexity. The Lorenz attractor (Fig. 1c) is a prime example of complex behaviour arising from feedbacks in dynamical systems. We used the code as provided by Moiseev (2022) with standard parameters to produce a time series of the Lorenz attractor. From its three variates, for clarity only the first one is shown and discussed, the results for jointly considering all three variates are similar. All synthetic time series consist of $nt = 30000$ time steps, and both for value binning and entropy binning ten bins were used. With a choice of $m = 3$, the range of recommended time slice widths is $sw = [30,1000]$ according to Eq. 4. In addition to the recommended range of time slice widths, we also included the two extremes values $sw = 1$ and $sw = 30000$ for demonstration purposes.

Time series d-f are hydro-meteorological observations taken from the CAMELS US data set (Newman et al., 2015). The first (Fig. 1d) is daily precipitation observations for the South Toe River, NC (short: STR) basin, the second (Fig. 1e) is the corresponding time series of daily streamflow observations. The basin size is 113.1 km², and precipitation mainly falls as rain (fraction of precipitation as snow is 8.5%). The third time series (Fig. 1f) also contains daily streamflow observations, but from the 111.5 km² Green River, MA (short: GR) basin, which is more snow-dominated (fraction of precipitation as snow is 22.2 %). We chose the time series for the following reasons: Comparing precipitation and streamflow series from the same basin (STR) allows analysing the effect of the rainfall-runoff transformation process on uncertainty and complexity. Here we expect that a basin - by spatio-temporal convolution of precipitation - mainly reduces precipitation variability, and with it uncertainty and complexity. Comparing streamflow from two basins with different levels of snow influence (STR and GR) allows analysing the effect of snow processes on uncertainty and complexity. Here we expect that the carryover effect of snow accumulation, and the influence of an independent additional driver of hydrological dynamics – radiation – should increase both uncertainty and complexity. All hydro-meteorological time series contain 12418 daily observations from 1 October 1980 – 30 September 2014 (34 years). As for the synthetic time series, we also used ten bins to resolve both the range of values and the range of entropies. However, we used a different time slicing scheme to reflect standard ways of time-aggregation of real-world data. In particular, we used the set of $sw = \{1,7,14,21,30,60,91,182,365,730,12418\}$ days, which corresponds to 1 day, 1-3 weeks, 1-6 months, 1-2 years, and the entire 34-year period. Please note that for a choice of $m = 3$, the range of recommended time slice widths is $sw = [30,414]$ days according to Eq. 4. Results for time slices outside of this range should therefore be treated with caution. We included them nevertheless for a more complete assessment of the time series.

For convenience we normalized all six time series to a [0,1] value range and then calculated uncertainty and complexity according to Eqs. (2) and (3).

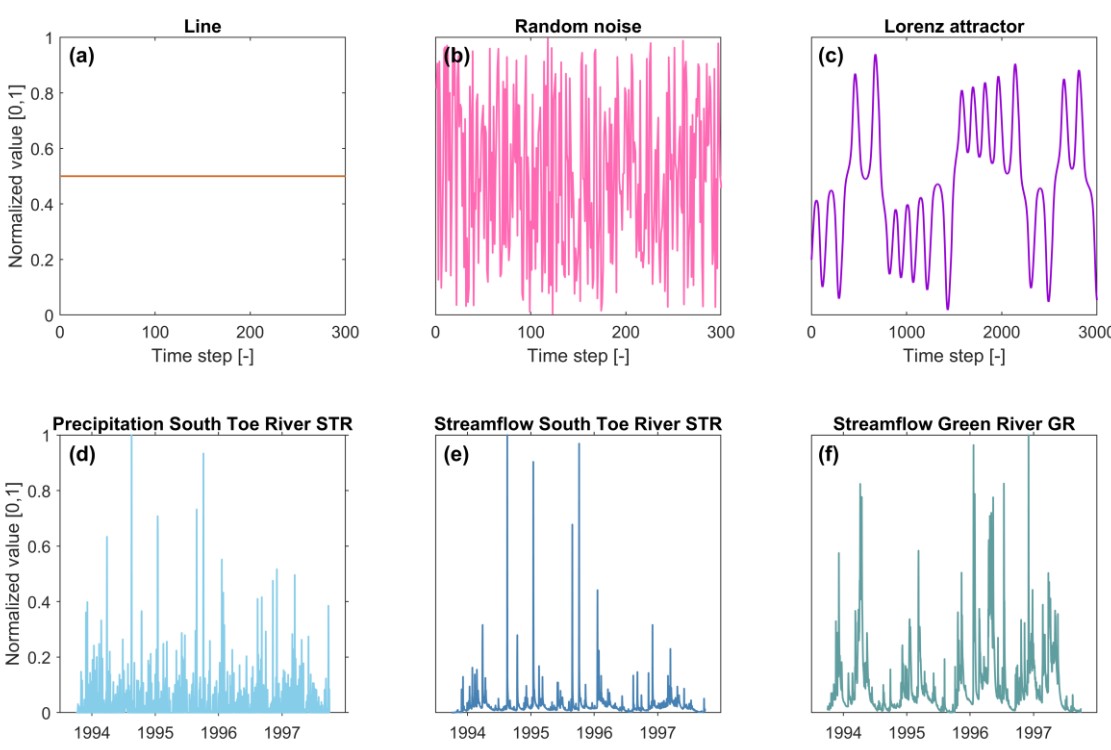

**Figure 1**. Synthetic and hydro-meteorological time series used for demonstration of the c-u-curve. Time series for subplots (a-c) comprise 30000 time steps; for clarity only 300 (subplots (a-b)) and 3000 (subplot (c)) time steps are shown. Time series for subplots (d-f) comprise 12418 daily time steps (34 years); for clarity only four years (1 October 1993 – 30 September 1997) are shown. All values are normalized to [0,1] value range. Further details on the time series are provided in the text.

**3.2 Results and discussion**

In this section, we present and discuss the c-u-curves of all six time series. We start by discussing the three artificial time series, followed by the three hydro-meteorological time series. All c-u-curves are shown in Fig. 2, and their key characteristics are summarized in Table 1. For clarity, Fig. 3 additionally shows only the hydro-meteorological time series in a sub region of Fig. 2. For further illustration, selected histograms of time series streamflow GR are shown in Appendix A.


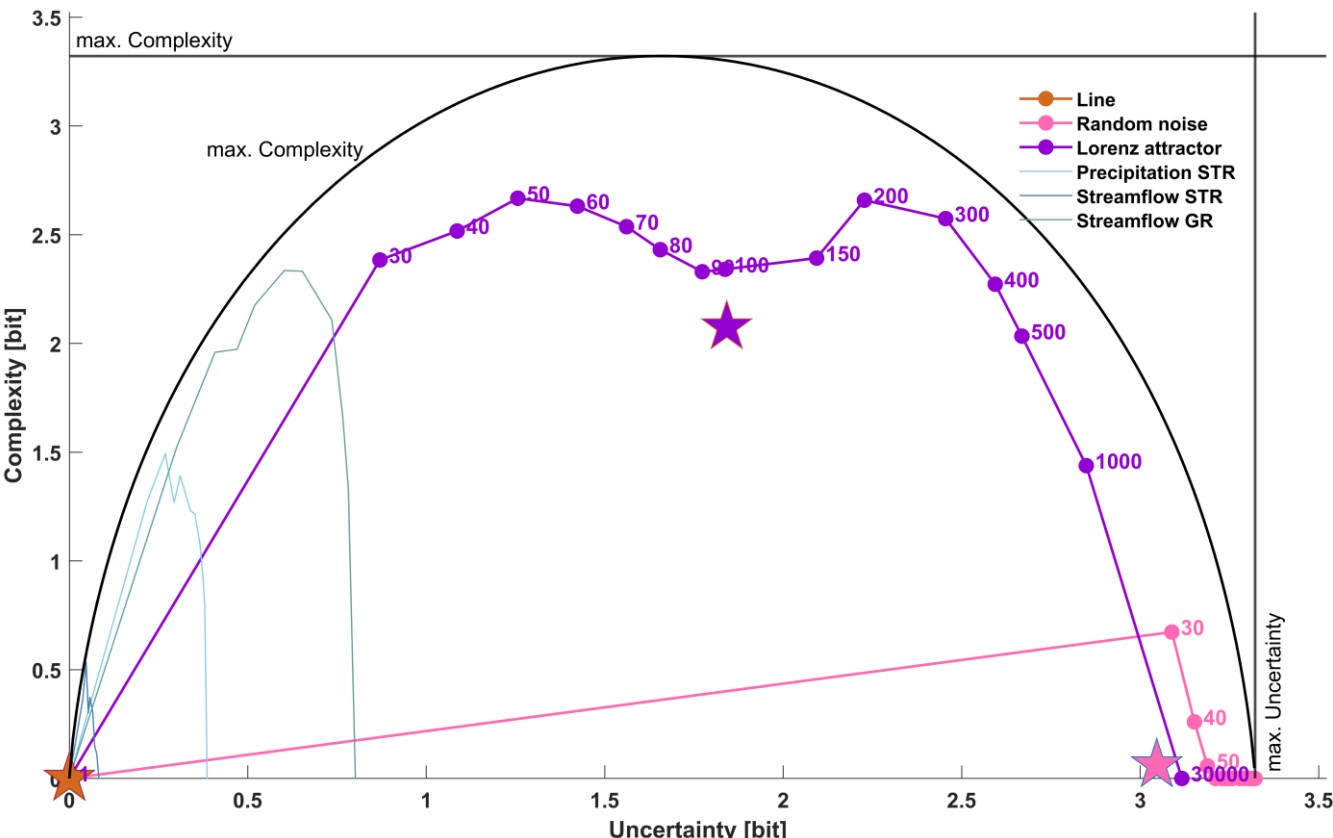

**Figure 2**. C-u-curves for synthetic (dotted) and hydro-meteorological (no marker) time series as shown in Fig. 1. Time series length is 30000 for the synthetic data and 12418 for the hydro-meteorological data. The number of value bins and entropy bins is ten, maximum uncertainty limit and maximum complexity limit is at $log_2(10) = 3.32$ bits. The black arch shows the maximum complexity limit as a
function of uncertainty. For the synthetic series, dot labels indicate the time slice width *sw* used to calculate uncertainty and complexity, and the pentagram positions indicate mean uncertainty and mean complexity across all chosen time slicing schemes. The hydro-meteorological series are included to indicate their position within the full range of uncertainty and complexity; their details are shown in Fig. 3. For interpretations of the axes units "bit", see Sect. 2.1. The lines connecting individual c-u points was included for better visibility, and should not be interpreted as an indication of guaranteed continuity of a c-u-curve.


   The overall shape of each c-u-curve contains key characteristics of the underlying time series. We start by discussing the c-u-plot of the *straight line* in Fig. 2: It shows – as expected - the simplest behaviour: For all time-slicing schemes, both within-slice and across-slices variability is zero, i.e. the series displays zero uncertainty and complexity throughout (all dots are stacked at the origin). As a consequence, mean uncertainty and complexity across all time-slicing schemes (indicated by the
brown pentagram in the plot and listed in Table 1) is also zero.

   The *random noise* series in Fig. 2 on the contrary displays very high uncertainty and low complexity for most of the time slicing schemes (most dots are stacked in the lower right corner of the plot), and only for many but narrow time slices of 50, 40 and 30 values per slice does complexity assume non-zero values. This can be attributed to random effects in small samples, where purely by chance both highly and hardly variable samples can occur, thus creating a wide range of time slice

entropies, resulting in apparent non-zero complexity. For wider slices, the larger sample size leads to more similarly
distributed samples, resulting in a narrow range of time slice entropies and hence low complexity. Overall, mean uncertainty
is very high and mean complexity is very low (position of the pink pentagram in Fig. 2 and values in Table 1), which is what
we expected from random noise..

The *Lorenz attractor* in Fig. 2 reveals a more diverse behaviour across the range of time slicing schemes. We start discussing
it for the case of $sw = 30000$, i.e. when a single time slice covers the entire time series. As described in the general
properties, for this case uncertainty is always at its maximum and equals the entropy of the time series, and complexity is
zero, because only a single entropy value populates the entropy distribution. The actual uncertainty value (3.11 bits), or its
distance from the upper limit of uncertainty ($3.11/3.32 = 94\%$), is a key characteristic of the time series and expresses its
overall variability. Decreasing the time slice width $sw$ decreases within-slice variability (uncertainty). Also, it provides the
potential for nonzero complexity as more and more entropy values populate the entropy distribution. For the curve shown in
Fig. 2, complexity continuously increases and reaches its first maximum value of 2.66 bits (or $2.66/3.32 = 80\%$) for $sw =$
200 and at 2.22 bits of uncertainty. This point is another key characteristic of a c-u-curve, indicating at which temporal
aggregation the across-slices variability is highest. Further decreasing slice width first leads to a decrease and then another
increase in complexity until a second maximum of 2.66 bits is reached at $sw = 50$ (see values in Table 1). Afterwards,
complexity and uncertainty decrease to zero for $sw = 1$, which is a general property of any c-u-curve (see discussion of
general properties above). Taking the uncertainty and complexity mean across all time slices summarizes the c-u-curve in a
single point (purple pentagram in Fig. 2, values in Table 1). For the Lorenz attractor, it reveals medium average uncertainty,
and high average complexity. In fact, the overall shape of the c-u-curve is close to the upper complexity limit reachable at a
given uncertainty (shown in the plot as a black arch).This is in accordance with expectations, as the Lorenz attractor is
known for exhibiting complex behaviour on many time scales. Interestingly, apart from revealing its generally complex
behaviour, the c-u-curve also reveals at which particular time slice width complexity of the Lorenz attractor is at a
maximum. This can be interpreted as a "characteristic time scale" of the time series.

**Table 1**. Key characteristics of the c-u-curves for both the synthetic and the hydro-meteorological time series.

| Time series | Uncertainty (bit) | | Complexity (bit) | | Characteristic time scale (days)[a] |
|---|---|---|---|---|---|
| | max | mean | max | mean | |
| Line | 0 | 0 | 0 | 0 | n.a. |
| Random noise | 3.32 | 3.04 | 0.67 | 0.06 | 30 |
| Lorenz attractor | 3.11 | 1.84 | 2.66 | 2.07 | 50, 200 |
| Precipitation STR | 0.38 | 0.30 | 1.49 | 0.88 | 14 |
| Streamflow STR | 0.09 | 0.06 | 0.53 | 0.17 | 14 |
| Streamflow GR | 0.80 | 0.57 | 2.33 | 1.45 | 60 |

[a] width of time slice at which maximum complexity occurred.

Next, we discuss the c-u-curves of the hydro-meteorological time series. In Fig. 2, they are indicated by the lines without
markers. It is immediately obvious that they all possess low uncertainty, much lower than the theoretical maximum
(indicated by the vertical "max. Uncertainty" limit) and the random noise, and also lower than the Lorenz attractor. This is in
accordance with our expectations, and a consequence of the typically high temporal autocorrelation of hydro-meteorological
time series, which clearly separates them from purely random time series. For a better view of detail, we re-plotted the
hydro-meteorological time series in a sub region of the uncertainty limits in Fig. 3, which we will refer to in the following.
Despite the generally low uncertainties, the *precipitation STR* time series in Fig. 3 displays considerable complexity
(indicated overall by the c-u-curves being close to the upper complexity limit, and for mean complexity by the relatively

high pentagram position), which can be explained by the existence of meteorological regimes with different levels of precipitation variability, such as dry periods (low variability), periods with alternating dry and wet periods (high variability), and wet times with diverse precipitation amounts (high variability). The highest complexity occurs for a time slice width of $sw = 14$ days, indicating that the greatest variability of within-slice precipitation variability occurs for two-week periods.

Interestingly, the corresponding *streamflow STR* time series displays much lower mean and maximum values (see Table 1)
for both uncertainty (within-slice variability) and complexity (across-slices variability). This is in accordance with the general hydrological understanding that in the absence of major carryover mechanisms, rainfall-runoff transformation in catchments is mainly by aggregation and convolution, thus reducing the variability of the precipitation signal. It is noteworthy that while this harmonizing effect changes uncertainty and complexity means and maxima, it does not affect the characteristic time scale: For streamflow STR - just as for precipitation STR - it is two weeks. This suggests that
precipitation remains the main control of streamflow complexity, despite the processes involved in rainfall-runoff-transformation.

This is different for the second *streamflow GR* time series. Here, in addition to the above-mentioned rainfall-runoff transformation, precipitation is partly stored as snow and later released as streamflow by melting. The temporal pattern of snowmelt is not only governed by snow availability, i.e. the precipitation regime, but also energy availability, i.e. the long-
term radiation and temperature regime. Such additional, independent controls of hydrological function can add uncertainty and complexity to streamflow production. Compared to streamflow STR, both uncertainty and complexity are indeed much larger in terms of mean and maximum values, they are even larger than the corresponding values for precipitation STR (compare pentagram positions in Fig. 3 and values in Table 1). The characteristic time scale of streamflow GR is at 2-3 months (60-91 days). This is considerably longer than for streamflow STR, and can be explained by the carryover effect of
snow accumulation and snowmelt acting at time scales in the order of months rather than days or weeks. For further illustration of the c-u-curve method, selected histograms for streamflow GR are shown in Appendix A.

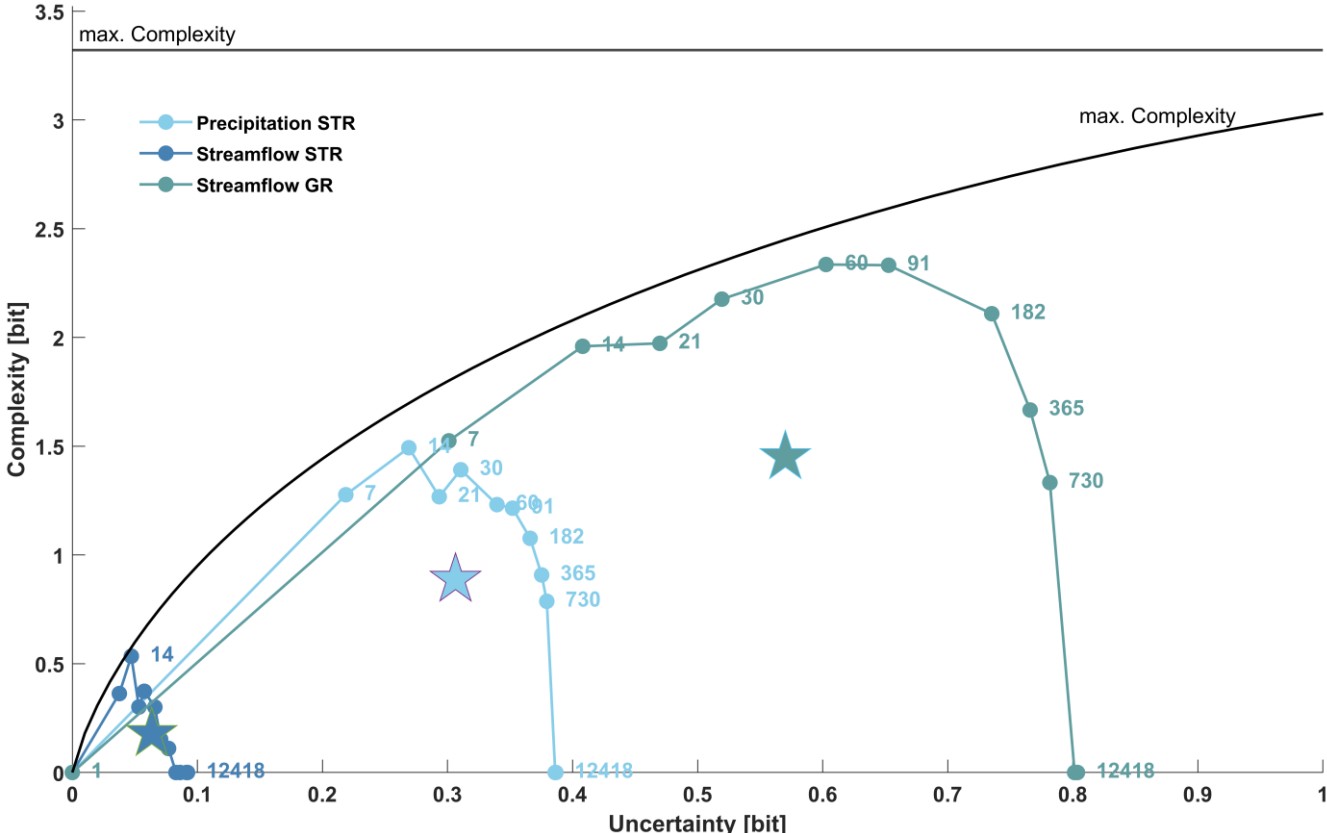

**Figure 3.** C-u-curves for all hydro-meteorological time series as shown in Fig. 1d-f. All time series comprise 12418 time steps, the number
of value bins and entropy bins is ten, maximum uncertainty limit and maximum complexity limit is at $log_2(10) = 3.32$ bits. The black

arch shows the maximum complexity limit as a function of uncertainty. Note that for better display of details this is a horizontally zoomed-in version of Fig. 2. Dot labels indicate the time slice width *sw* used to calculate uncertainty and complexity. The pentagram positions indicate mean uncertainty and mean complexity across all chosen time slicing schemes. For interpretations of the axes units "bit", see Sect. 2.1. The lines connecting individual c-u points were included for better visibility, and should not be interpreted as an indication of guaranteed continuity of a c-u-curve.

## 4 Summary and conclusions

In this paper we presented a method to analyse and classify dynamical systems by the two key features uncertainty and complexity. After dividing the time series into a set of time slices, the Shannon information entropy is calculated for the data in each time slice. *Uncertainty* is then calculated as the mean entropy of all time slices, *complexity* as the entropy of all entropy values. Complexity thus expresses "uncertainty about uncertainty" in the time series. Calculating and plotting uncertainty and complexity for many time slicing schemes yields the *c-u-curve,* with key characteristics mean and maximum uncertainty, mean and maximum complexity, and the characteristic time scale of the time series. The latter is defined as the time slice width at which maximum complexity occurs.

The c-u-curve method has several useful properties: Independence from the units of the data (both uncertainty and complexity are expressed in bit), existence of upper and lower bounds for both uncertainty and complexity as a function of the chosen data resolution, and bounded behaviour when approaching upper and lower limits of time-slicing: For a single time-slice containing all data, uncertainty equals the time series entropy and complexity is zero, for time-slices containing single values both uncertainty and complexity are zero. The c-u-curve method is applicable to single- and multivariate data sets, and to deterministic and probabilistic value representations (ensemble data sets), making it suitable for a wide range of tasks and systems. The main limitation of the method arises from the requirement of sufficiently populating distributions, which sets bounds to both the minimum and maximum width of time slices.

We provided a proof-of-concept at the example of six time series, three of them artificial, three of them from hydro-meteorological observations. The artificial time series (straight line, random noise, Lorenz attractor) were chosen for their very different, exemplary and well-known behaviour, and with the goal to demonstrate that the c-u-curve successfully reveals this behaviour, i.e. to demonstrate the general applicability of the method across a wide range of time series types. The observed time series (precipitation and streamflow from a mainly rainfall-dominated basin, and streamflow from a basin where additionally snow processes influence the hydrological function) were chosen with the goal to demonstrate that the c-u-curve method reveals characteristics of real-world time series that are in accordance with general knowledge of hydrological system functioning. For all time series, we could show that the c-u-curve properties were distinctly different among the time series – which indicates that the method has discriminative capabilities useful for system classification -, and that the properties are in accordance with expectations based on system understanding – which indicates that the method captures relevant time series properties and expresses them in terms of uncertainty and complexity -.

While the range of applications presented in this paper is small, and mainly intended as a proof-of-concept, the results encourage further studies. Particularly for hydro-meteorological applications, we suggest that the c-u-curve method can be used for hydrological classification, as objective function in hydrological model training, and for hydrological system analysis. For classification, we suggest using large hydrometeorological data sets such as from Addor et al. (2017) or Kuentz et al. (2017) for analysing whether the c-u-curve distinguishes among catchments with known differences, such as groundwater and interflow dominated, pristine and regulated, snow-free and snow-influenced, arid and humid. In the same context, classifications by the c-u-curve can be compared to existing hydrological classifiers and signatures (such as the flow-duration curve and others as discussed in Jehn et al., 2020; Addor et al., 2018; Kuentz et al., 2017) in terms of classification similarity and strength. The clear differences of c-u-curve properties between the two streamflow time series investigated in this paper encourage further research in this direction. In terms of hydrological model training, we  suggest

that the c-u-curve and its characteristic values can be used as an additional objective function: While standard hydrological objective functions such as Nash-Sutcliffe efficiency guide models towards point-by-point agreement of model output and observations, c-u-curve characteristics can guide models towards correct representations of short- and long-term variability patterns. Supported by the (dis-)similarities of the c-u-curve properties of the precipitation and streamflow time series presented in this paper, we also suggest that by analysing and comparing c-u-curve properties of input, internal states and output of hydrological systems, valuable insights about the functioning of these systems can be gained, e.g. if they in- or decrease uncertainty and complexity of the signals propagating through them. Further work on these topics is in progress. Finally, we propose the combination of the multiscale entropy (MSE) and the c-u-curve approach as discussed in Sect. 2.3 as a very promising avenue for future work.

**Appendix A: Histograms for time series streamflow GR**

As an illustration how time series values within a time slice translate into histograms and entropy values, we show, for streamflow GR, in Fig. A1 the streamflow hydrographs and corresponding histograms for three time slices. All time slices have a width of 60 days, which is the slice width for which the series shows highest complexity (compare Table 1 and Fig. 3). Overall, the time series (12418 time steps) splits into 12418/60 = 206 time slices. For each slice, we calculated entropy and selected three interesting ones: One with the smallest of all entropy values (0 bits), one with the highest of all entropy values (2.27 bits), and one with an entropy of 0.61 bits, which is close to the overall mean entropy of 0.60 bits of all 206 time slices ("uncertainty"). The normalized time series of the three 60-day slices are shown in Figs. A1a,b,c, the corresponding histograms are shown in Figs. A1d,e,f. As can be seen from Fig. A1a-c, for streamflow GR the possible range of variability within 60-day time slices is quite high, ranging from almost uniform flow (Fig. A1a) to time slices including very variable flow with both high and low flow conditions (Fig. A1c). This is summarized in Fig. A2, which shows the histogram of all 206 entropy values. Its entropy ("complexity") is 2.33 bits (compare Table 1).

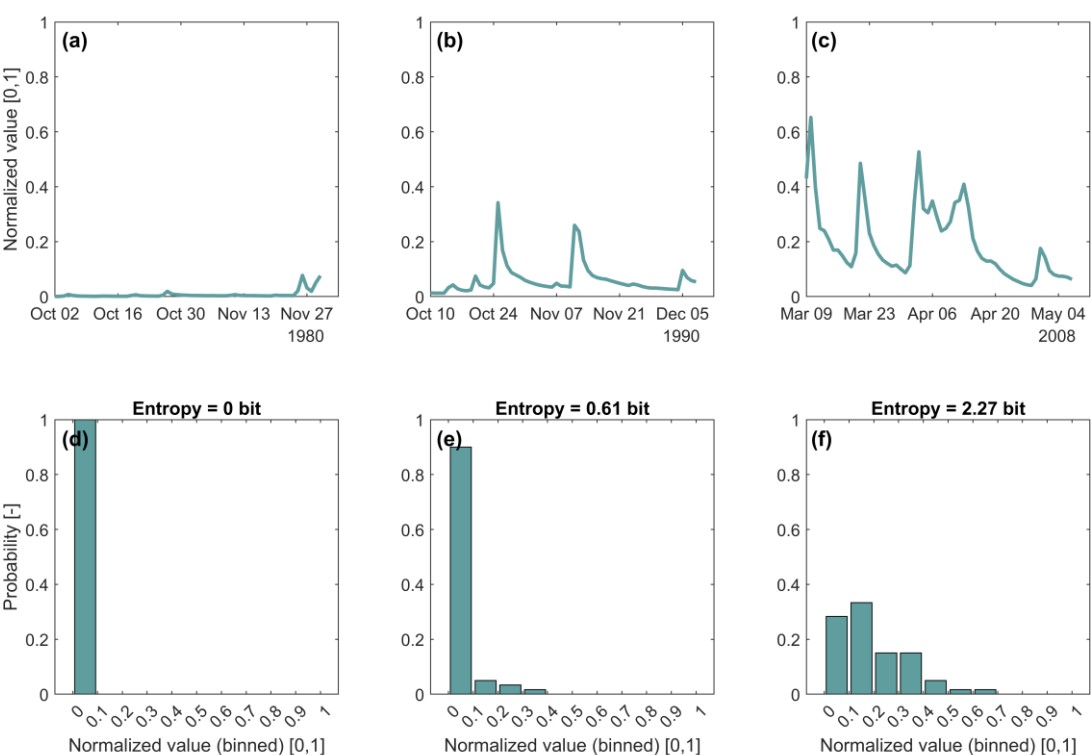

**Figure A1**. Normalized streamflow hydrographs and corresponding histograms of three time slices from time series streamflow GR. Each time slice comprises 60 days. For the histograms, the value range of the normalized streamflow was split into ten bins of uniform width.

Subplots a) and d): time slice 2 October – 30 November 1980, entropy = 0 bits; subplots b) and e): time slice 10 October – 8 December 1990, entropy = 0.61 bits; subplots c) and f): time slice 9 March – 7 May 2008, entropy = 2.27 bits.

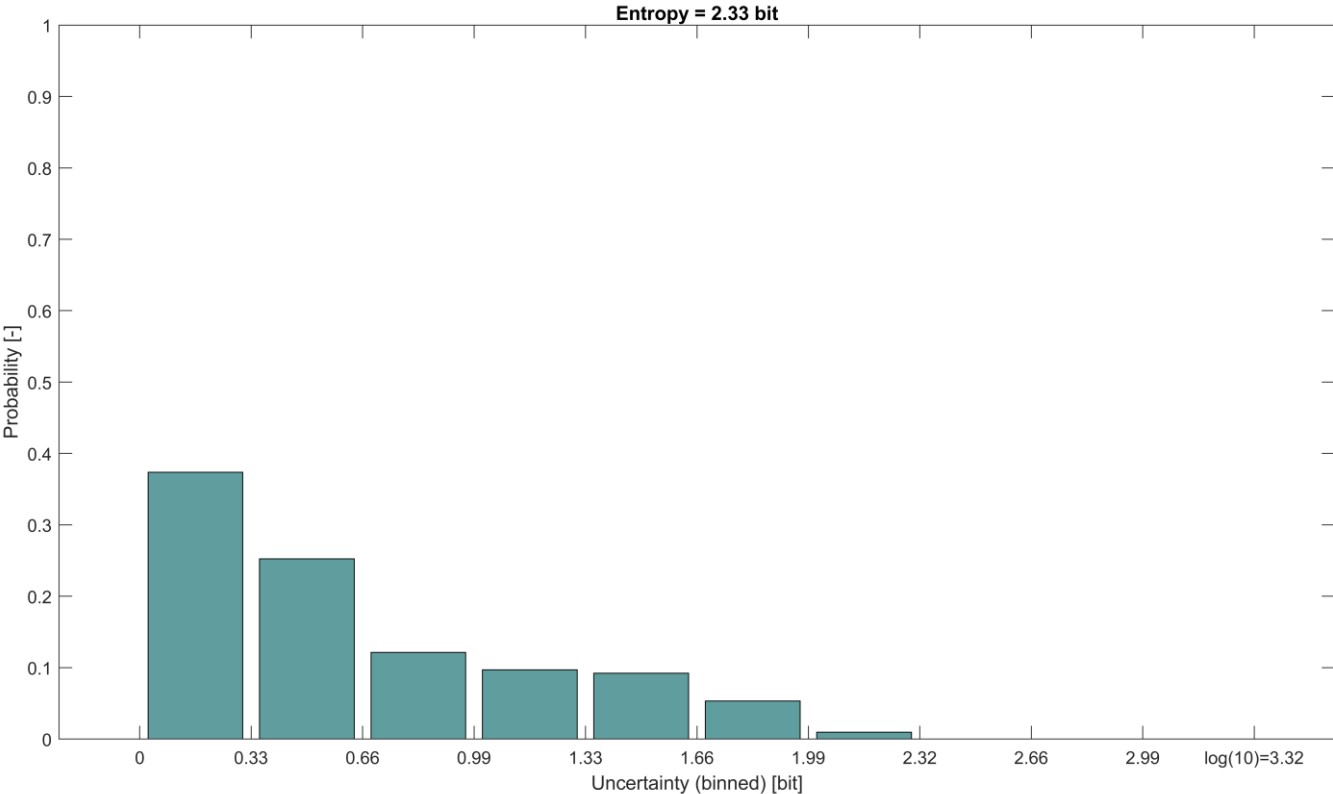


**Figure A2**. Histogram of entropies from normalized time series streamflow GR split into 206 time slices, each with a width of 60 days. Entropy for each time slice was calculated from histograms (see Fig. A1). For the histogram, the possible range of entropy values ([0, $log_2(10) = 3.32$] bits was split into ten bins of uniform width. The entropy of the histogram "complexity") is 2.33 bits (compare Table 1).

**Appendix B: Proof for the existence of an upper bound of the c-u-curve**

For the convenience of the reader, we repeat Theorem 5.12 from Conrad (2022) here and some related explanation in slightly modified and shortened form, but for the full proof, for brevity the reader is referred to the original publication. In the following, $S = \{s_1, \dots s_n\}$ refers to a finite set of discrete, distinguishable states of a (physical) system, with corresponding energy states $\{E_1, \dots, E_n\}$ and probabilities $\{p_1, \dots p_n\}$ of the system to be in a particular state. For each probability

distribution $p$ on $S$, the corresponding expected value of $E$ is given by Eq. B1.

$$\bar{E} = \sum p_j \cdot E_j \tag{B1}$$

This number is between $\min E_j$ and $\max E_j$. For a chosen (a priori known) value of $\bar{E}$, the goal is to find the probability distribution $q$ with the given $\bar{E}$ and maximum entropy. For the general case when $q$ is not a uniform distribution, Theorem 5.12 provides a semi-analytical solution.

**Theorem 5.12**. If the $E_j$'s are not all equal, then for each $\bar{E}$ between $\min E_j$ and $\max E_j$ , there is a unique probability

distribution $q$ on $\{s_1, \dots s_n\}$ satisfying the condition $\sum q_j E_j = \bar{E}$ and having maximum entropy. It is given by the formula

$$q_j = \frac{e^{-\beta \cdot E_j}}{\sum_{i=1}^{n} e^{-\beta \cdot E_i}} \tag{B2}$$

for a unique extended real number $\beta$ in $[-\infty, \infty]$ that depends on $\bar{E}$. In particular, $\beta = -\infty$ corresponds to $\bar{E} = \max E_j$, $\beta = \infty$ corresponds to $\bar{E} = \min E_j$, and $\beta = 0$ (the uniform distribution) corresponds to the arithmetic mean $\bar{E} = \left(\sum E_j\right)/n$, so $\beta > 0$ when $\bar{E} < \left(\sum E_j\right)/n$ and $\beta < 0$ when $\bar{E} > \left(\sum E_j\right)/n$. The value of $\beta$ can be numerically approximated with an iterative algorithmic recipe and Eqs. B1 and B2 (see example 5.14 in Conrad, 2022).


*Code availability.* The code used to conduct all analyses in this paper is publicly available at https://doi.org/10.5281/zenodo.7276917 (Ehret, 2022).

*Data availability.* All data used to conduct the analyses in this paper and the result files are publicly available at
https://doi.org/10.5281/zenodo.7276917 (Ehret, 2022).

*Author contributions.* UE developed the c-u-curve method and wrote all related code. UE and PD designed the study together and wrote the manuscript together.

*Competing interests.* The authors declare that they have no conflict of interest.

*Acknowledgements.* We gratefully acknowledge support by the Deutsche Forschungsgemeinschaft (DFG) and the Open Access Publishing Fund of the Karlsruhe Institute of Technology (KIT). We thank Philipp Reiser from University of Stuttgart for pointing us to Conrad (2022). The second author is grateful to Department of Science and Technology (DST),
Government of India, for providing him DST INSPIRE Faculty Fellowship in 2023 (Faculty Registration No. : IFA22-EAS 114, Application Number: DST/INSPIRE/04/2022/001952).

*Financial support.* The article processing charges for this open access publication were covered by a Research Centre of the Helmholtz Association.

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
