# Peer review of "Technical note: c-u-curve: A method to analyse, classify and compare dynamical systems by uncertainty and complexity"

_Hydrology and Earth System Sciences, 2022_

## Author Comment (AC1)

**hess-2022-16 Responses to comments by referee #1 (Jasper Vrugt)**

Dear Editor, dear Referee,

We thank the first referee, Jasper Vrugt, for the very detailed and constructive review of our manuscript. Working on the replies triggered a lot of very valuable additional insights, and they will definitely help us to better communicate our research. We will in the following reply to the comments point by point. The Referee comments are in blue.

**Comment 1:** Summary: The authors resort to information theory and present the so-called c-u-curve to describe/quantify characteristic properties of hydrometeorological data. This c-u-curve displays graphically the relationship between what authors refer to as system uncertainty and complexity. This information is thought to be expressed and/or contained within spatial and/or temporal measurements of the data generating process of interest. **System uncertainty** is defined as the mean Shannon information entropy of many different time slices (time windows). The authors define **system complexity** as the '…uncertainty about uncertainty' (P1, Line 11) and express this quantitatively as the entropy of the entropies of all time slices. As the two metrics depend strongly on the temporal extent (width) of the time window, the authors repeat their analysis for many different slice sizes. The c-u-curve is a graphical depiction of the relationship between the so-obtained system uncertainty (x-axis) and system complexity (y-axis), both of which have units of bits. The authors illustrate this idea by application to six different signals (time series), including simulated data of a (i) deterministic (horizontal line), (ii) random (normally distributed variates) and (iii) chaotic system (Lorenz attractor) and measured time series of (iv) precipitation and (v,vi) catchment discharge of the South Toe and Green Rivers in the United States. The authors conclude that the c-u-curve can be used to analyze, classify and compare dynamical systems.

Evaluation: The manuscript discusses an important topic in hydrology and complex systems analysis in general, namely the characterization of the dimensionality and complexity of dynamical systems. I enjoyed reading this manuscript. The document is well written and relatively easy to understand. Rationales and ideas are clearly presented. The six case studies demonstrate/showcase the potential use of the c-u-curve, inform readers about the methodology and how to interpret its results. I applaud the authors for their work, which I believe is very interesting. I do have serious concerns however about the mathematical and literature underpinning of the methodology, and the robustness and convergence properties of the c-u-curve. Based on these comments, I recommend a major revision.

Reply 1: We are glad that the referee finds our work interesting and relevant. The main points of concern raised by the referee (mathematical and literature underpinning of the method, robustness and convergence properties of the method) will be addressed below in the corresponding specific detail comments.

**Comment 2**: General comment. The authors decided to present their work in the form of a technical note. This is an efficient way to rapidly disseminate new ideas. But technical notes have strict length requirements which can make it difficult to address all important aspects of the work presented. The ideas presented are very interesting, yet a full paper may do more justification to the ideas and work presented. I have several questions about the methodology, which I think should be addressed before readers can judge that what is presented is a substantial and/or important advance in our ability to analyse, classify and compare dynamical systems. Note that in my review below I use the word 'signal' for a measured or simulated time series of some quantity of interest. I also use the word 'paper' in reference to this technical note. This word is conveniently used and should certainly not imply that I was expecting a much longer manuscript.

Reply 2: Initially we indeed thought about presenting the c-u-curve method in a full scientific paper, with the method description and a range of applications, including:

- Hydrological classification: Use data from hydrologically distinctly different catchments such as groundwater dominated, interflow dominated, dominated by reservoir operation, arid, humid and snow-influenced catchments from large data sets (e.g. Addor et al. 2017, Kuentz et al., 2017) and see if and how these differences are reflected by c-u-curve properties).
- Comparison to existing hydrological classifiers and signatures (such as those discussed in Jehn et al., 2020; Addor et al. 2018; Kuentz et al., 2017) at the example of large data sets (e.g. Addor et al., 2017). This includes evaluating the classification power of c-u-curve and the evaluation how similar or dissimilar its classifications are to those from existing classifiers.
- Use for model improvement: Test if c-u-curve characteristics can be used for targeted model improvement, either as an objective function for parameter identification during model calibration, or as a signature which may point to model structural deficiencies
- System analysis: Compare c-u-curve characteristics for input, internal states and output of hydrological systems to analyse system behaviour: Are uncertainty and complexity increasing or decreasing on the way through the system?

Looking at this list we decided it will be too much material for a single paper to introduce the method and provide all these use-cases. So instead we decided to introduce the c-u-curve method in a compact manner in a technical note, along with a few examples illustrating its properties and behaviour, and presenting the other aspects in a follow-up scientific paper. We suggest that this is the best compromise for rapid yet thorough presentation of the method. To clarify this to the reader, we suggest adding in a revised version of the manuscript a sentence to the last paragraph of section 4 ('summary and conclusions'), explaining possible further avenues of research along the above bullet-point list.

**Comment 3:** Section 2.1: The authors resort to information theory to analyze the temporal properties of the signal of interest. They coin two measures of system functioning/behavior, namely system uncertainty and system complexity and provide a mathematical definition for both that are subsequently used to construct the c-u-curves in the analysis. No references and/or background is provided about their definition. There is a large body of literature on complex systems and imagine that others have defined similar metrics to classify, describe and characterize time series data. This begs the questions whether the two criteria stand completely on their own and if earlier attempts have been made to analyze time series data in a similar fashion? I think the paper would be considerably stronger if the authors can relate their work to previous published work. Have other definitions of these criteria appeared in the complex systems literature?

Reply 3: Uncertainty and complexity have no single agreed-upon definition in hydrology and in the earth sciences in general. Instead, many authors have suggested different definitions and methods to quantify them (see references cited in line 36, and lines 48-50 in the manuscript). But to the best of our knowledge, we are not aware of an approach identical to the c-u-curve method, both in the hydrological and in the complex systems literature. We searched especially the hydrological literature to find comparable approaches based on information measures. Here, the work by Pachepsky, Hauhs and Lange stands out, and we included it in the introduction (line 51). From the field of physics, LopezRuiz et al. (1995) and Feldman and Crutchfield (1998) suggest similar, but not identical approaches. Nevertheless, we followed the referee's suggestion and repeated the literature search. In a revised version of the manuscript, we suggest including (and discussing) into the introduction the following references:

- About complex systems: Ladyman et al. (2013), Lloyd (2001), LopezRuiz et al. (1995), Feldman and Crutchfield (1998)

- About uncertainty in hydrological modelling (predictive uncertainty, model structural and parameter uncertainty): Vrugt et al. (2003), Liu and Gupta (2007), Vrugt et al. (2009).

**Comment 4**: Section 2.1: Related to my previous comment, what is wrong with specifying system complexity as the temporal variance of the Shannon entropies? Then you the first moment (mean) as measure of system uncertainty and the second moment (variance) as measure of system complexity. I am sure the authors have thought about this. In the present paper I just miss rationales and arguments so as to why their definitions are appropriate – also in light of past work done in the literature on this topic.

Reply 4: Yes, this makes a lot of sense from the perspective of describing distribution in terms of their moments (first, second, etc.). However, we think there are good reasons for expressing variability in terms of entropy:

- Independence from units: Expressing the joint variability of a multidimensional data set within a time slice is difficult to achieve by variance, as the variance of each variable dimension comes with its own unit, which makes direct combination impossible. A way out is to standardize variances (e.g. by the variation coefficient) before combination. As entropy operates on probabilities, the units of the variables play no role, which we think is an advantage when working with multivariate data sets, which is a key use case for the c-u-curve method.
- Consistency of the method: if we express variability of the distribution within a time slice by entropy for the reasons given in the previous bullet point, we think that it is a natural choice to express variability of the variabilities also by entropy. Thus, variability is always expressed in the same units, which increases interpretability of the method. Also, by expressing both variabilities in terms of entropy, useful and comparable upper bounds can be established, which would not be the case if variance is used for expressing complexity.
- Interpretability: Entropy has the (in our eyes) very intuitive interpretation of 'number of binary Yes/No questions to ask to move from a prior to a posterior state of knowledge' (e.g. guessing a value coming from a known distribution). Variance lacks this straightforward interpretation.
- Robustness: Variance is dominated by values far away from the mean and is therefore sensitive to outliers in the data set. Entropy is dominated by frequent values, i.e. the centre of a distribution, which makes it less sensitive to outliers. While for extreme value-statistics, where the tails of a distribution are of special interest, variance is a good choice, we think that for a characterization of the overall variability of a data set, entropy is a better choice.

We suggest adding in a revised version of the manuscript to section 2.1 (method description) a brief explanation about why we express uncertainty and complexity by entropies rather than variance along the lines of the above bullet points.

**Comment 5:** Section 2.1: In their analysis of the signal, the authors coined the words system complexity and system uncertainty. I do not think these labels are accurate. System uncertainty and system complexity refer to the system as a whole – and should, in principle, not depend on which variable of the system is observed. They should be invariant properties (unless the system experiences change). Instead, what the authors determine is the uncertainty and complexity of the signal only. Thus, I think it is more accurate to use the words signal uncertainty and signal complexity. Indeed, I expect you will get different c-u-curves for different signals of system behavior. If we take hydrology as example, then soil-moisture will likely give a different c-u-curve than a time series of groundwater table depths and this curve will be different from its counterpart of the discharge. Certainly, I would argue that a single component of system behavior is insufficient to characterize the complexity and uncertainty of the system as whole.

Reply 5: We agree. In a revised version of the manuscript, we will use the terms 'signal uncertainty' and 'signal complexity' instead of 'system uncertainty' and 'system complexity'. We will also add a related sentence describing that we will only be able to capture true system uncertainty and complexity if we include all of the system's state variables into the analysis, which is impossible for natural systems (basically repeating the referee's argument). We will also mention that an advantage of the c-u-curve method is that it allows joint treatment of all available system signals, such that we can (with increasing availability of different signals), approach system uncertainty and complexity.

**Comment 6:** Section 2.1: The choice of the number of time slices and their spatial extent; I'll call this the temporal discretization of the signal; play a crucial role in the analysis. Without derivation and much explanation at all the authors introduce Equation (4) which provides a lower and upper bound for the width of the time slices. How is this equation derived? Is this a rule of thumb? The lack of a theoretical underpinning is a concern. It may be productive to have a look at Sturges method (or for that matter Scott's method or Freedman-Diaconis) which provide a rule of thumb for the number of histogram bins that should be used for a given length of data. This may be used to improve the statistical underpinning of Equation (4).

Reply 6: Eq. (4) formalizes constraints on the range of suitable time slice widths as a consequence of the length of the time series (nt) and user choices on the binning resolution (nvv and neb) and a desired average binning population (m). The latter three choices can be made based on the methods mentioned by the referee (Sturges, 1926; Scott, 1979; Freedman and Diaconis, 1981) or others such as (Knuth, 2019; Pechlivanidis et al., 2016). However, while the three user choices include some subjectivity, Eq. 4 simply, and without subjectivity, express how from these choices upper and lower limits of time slices arise as a consequence of the antagonistic interplay of constraints for uncertainty calculation according to Eq. (2) (preferably wide – hence few - time slices) and complexity calculation according to Eq. (3) (preferably many – hence narrow – time slices). This is described in lines 94-104.

As this was apparently hard to understand, we suggest adding to a revised version of the manuscript in section 2.1 a sentence including possible methods for binning choice (including the references above) and an explanation that Eq. (4) itself is not subjective, but formalizes hard constraints based on subjective choices made by the user.

**Comment 7**: Section 2.1: Readers may be interested to see a few of the histograms that went into the computation of system uncertainty and system complexity. Do the histograms differ substantially from one time slice to the next? Do they have an overarching distribution? Skew, kurtosis, etc?

Reply 7: Good point. For illustration, we have chosen the time series "streamflow Green River GR", and calculated uncertainties for time the slice width "60 days". This is the slice width for which the series shows highest complexity (compare Table 1 and Figure 3 in the manuscript). Overall, the time series (12418 time steps) splits into 12418/60=206 slices. For each slice, we calculated entropy. From the 206 time slices, we selected three interesting ones: One with the smallest of all entropy values (0 bit), one with the highest of all entropy values (2.27 bit), and one with an entropy of 0.61 bit, which is close to the overall mean of the 206 values ("uncertainty"), 0.60 bit. The normalized time series of the three 60-day slices, and the corresponding histograms are shown in Fig. R1 ("R" for "figure in Reply", to distinguish them from Figures in the manuscript).

Comparing the time slice series with the corresponding histogram helps developing an intuition about how a time series maps into a histogram, and comparing the three histograms reveals the range of possible histograms. For the streamflow Green River data set and 60-day slices, the range is quite wide. This is summarized in Fig. R2, which shows the distribution of all 206 entropy values. Its entropy (="complexity") is 2.33 bit (compare Table 1 in the manuscript).

About the question of an overarching distribution: In principle, the histograms can take any shape between a Dirac and a uniform distribution. Each time slice distribution should to some degree be

influenced by the distribution of the overall data set the slices were taken from. E.g. if the overall distribution is highly skewed (e.g. rainfall), then chances are that the time slice histograms will also be skewed. A further investigation of this point is interesting, but we think it is beyond the scope of this manuscript.

[Figure]

Fig. R1

[Figure]

Fig. R2

We propose to add Fig. R1 and a short explanation into the Appendix of a revised version of the manuscript.

**Comment 8:** Figures 2 and 3: The authors assume that the c-u-curve is continuous, and connect the individual (u,c) pairs for individual time slices with a solid line. But is the curve continuous? Are there theoretical arguments from which one expects the curve to be continuous and not discrete?

Reply 8: Good point. We are not aware of theoretical arguments guaranteeing continuity. So in fact the most honest way to show the c-u-curve is by non-connected (u,c) dots. However, the general shape of the curve is best visualized by a line connecting the dots, which is the reason why we used it. To make this point clear, we suggest adding a sentence about 'no guarantee for continuity' to a new section 2.2 ('Properties') in a revised version of the manuscript. Further, we suggest adding similar short comments to the captions of Figs. 2 and 3.

To better illustrate the (dis-)continuity behaviour of the c-u-curve, we have repeated the calculations for time series "streamflow GR" (see Fig. 3 in the manuscript) for a large number of slice widths.

For Fig. 3 in the manuscript, we used the following slice widths:

- [1 7 14 21 30 60 91 182 365 730 6209 12418] days → overall 12 slice widths

For the new runs, we used the following slice widths:

- [1:1:100 110:10:300 350:50:3100 3145 6209 12418] days → overall 179 slice widths

In Fig. R3, the original (c,u) pairs are shown as red open circles (compare with Fig. 3 in the manuscript). The blue dots are from the new runs without moving window option, the green dots are from the new runs with the moving window option (please see Comment 19 for a description of the new "moving window" option).

[Figure]

Fig R3

In Fig. R3, it can be seen from the new runs that the c-u-curve behaves generally continuous, and that the red-circle c-u-pairs chosen for the manuscript reflect the overall shape of the curve. But again, there is no guarantee. E.g., there is a distinct discontinuity in the c-u-curve between slice width 16 days and 17 days (indicated by the arrows in Fig. R3), which could be caused by a single, exceptional event in the time series. However, this discontinuity should not be overemphasized, as for such narrow time slices (16 and 17 values in the slice, respectively), a robust population of the binned distribution (10 bin) is not guaranteed. According to Eq. 4, time slice widths between 30 and 1000 are recommended (see lines 106-107 in the manuscript).

We also repeated the calculations for all other time series. The results were very similar to those of "streamflow GR" shown above (not shown).

**Comment 9:** Figure 2: What happens to the Lorenz c-u-curve if we use windows (slices) of a size smaller than 30? Equation (4) suggests that such value is not recommended, but what happens to the curve itself? Does the curve oscillate close to the origin?

Reply 9: As in our reply to comment 8, we repeated the c-u-curve calculations for the Lorenz system, but with more slice widths. For Fig. 2 in the manuscript, we used the following slice widths:

- [1 30:10:90 100:50:200 300:100:500 1000 30000] → overall 16 slice widths

For the new runs, we used the following slice widths:

- [1:1:100 110:10:300 350:50:500 1000 30000] → overall 126 slice widths

The results are shown in Fig. R4. As before, the original (c,u) pairs are shown as red open circles (compare with Fig. 2 in the manuscript). The blue dots are from the new runs without moving window option.

[Figure]

Fig. R4

Similar to the results from the "streamflow GR" series that we showed in the reply to comment 8, the overall shape of the c-u-curve is generally smooth, and there are no oscillations close to the origin.

**Comment 10:** Line 145: The authors use normalization of the signal to yield values between 0 and 1. This itself is inconsequential yet allows a fixed recipe for data types with very different magnitudes. Why do the authors not use a similar normalization in the time domain? This may help (or not) to

standardize the characterization of the width of the slices. The only variable left is then the number of data points.

Reply 10: We agree with the referee that the time axis could also be scaled to [0,1], resulting in normalized time slice widths. In fact this can be very useful to increase comparability of c-u-curves derived from time series of different length. However, normalizing time comes at the cost of losing interpretability. E.g. for the observed time series 'streamflow GR' in Fig. 3, we think it is more helpful to see that maximum complexity occurs for time slices of 60 days rather than expressing this in [0,1] units as 60/12418=0.0048. We therefore prefer keeping the time scale in the original axis in the manuscript, but we suggest adding in a revised version a short remark to the sentence in line 146-147 that normalizing the time scale is also possible.

**Comment 11:** How does the c-u-curve respond to the frequency of measurement of the signal? For example, in the case of discharge, you can construct the curve for hourly, daily, weekly and monthly data (average flows) – do we see convergence of the c-u-curve to it counterpart of the horizontal line? I expect such convergence to be faster for timeaverage data points than for a signal made up of instantaneous measurements. This analysis is important as it will help establish convergence properties of the c-u-curve.

Reply 11: Very interesting question! To address it, we used the "streamflow GR" curve, and calculated block averages for various block sizes as suggested by the referee. In particular, we calculated averages for

- 2 days, 3 days, 4 days, 5 days, 6 days, 1 week, 2 weeks, 3 weeks, 1 month

For each averaged series, we calculated c-u-pairs for many different slice widths. The corresponding c-u-curves are shown in Fig. R5. The bold blue curve is for the original (non-averaged) series (compare Fig. 3 in the manuscript). Several interesting features are apparent in Fig. R5: First, all c-u-curves start at the origin, as to be expected. Second, maximum uncertainty (the lower right end of each c-u-curve), which is entropy of all values of the time series within a single large time slice generally increases with the size of the averaging block. This seems counterintuitive at first, as averaging should decrease variability, and with it entropy. The explanation is that by the averaging, the number of data points in the series changes (decreases), so it is not two series of the same number of time steps - one unsmoothed, one smoothed – that is compared, but rather it is two different series. Both series are normalized by their respective minimum and maximum values, and then entropy from the resulting 10-bin histogram is calculated. And apparently, daily values are less variable within their range of maxima and minima (most likely because there are many low-flow values), than e.g. monthly means, within their range of maxima and minima (most likely because seasonal variation causes a more uniform distribution of monthly values). Third, maximum complexity increases, then decrease with the size of the averaging block, for which we have no direct explanation, but we suspect it also has to do with the previously discussed effect.

In short, there is a clear effect of block-averaging on the c-u-curve. Exploring it would be very interesting, and probably fill a paper of its own, but beyond the scope of this manuscript. We therefore suggest adding to a revised version of the manuscript a sentence to a new section 2.2 ('Properties') explaining that the support (block size of summation or averaging) of the data has an influence on the results, and that this should be taken into account when comparing different c-u-curves (e.g. by comparing only curves from time series with the same support, which is also standard in conventional analysis: we usually do not compare daily temperature time series from one station with monthly averages from another station).

[Figure]

Fig. R5

**Comment 12:** How does the c-u-curve respond to a) numerical errors (in case of simulated signals) and b) system nonstationarity? This analysis is not difficult to do with simulated discharge data (for example using fixed step integration versus a variable time step or implicit solution) and will provide further insights into the method.

Reply 12: Again, very interesting questions!

Influence of a numerical error: If the error is **random**, and follows a certain known or assumed-to-be-known distribution around the true value (could also be a measurement error distribution in case of observation data), then our uncertainty about the true entropy of a time slice will be exactly the entropy of the error distribution, and total entropy of the time slice will be the sum of the within-slice entropy without the error (what we normally calculate) and the entropy of the error (entropy is additive for independent sources of information). As this added uncertainty will be the same for every time slice, mean entropy (uncertainty) will be increased by the same additive amount. As this means the distribution of entropies is simply shifted, but its shape remains unchanged, the entropy of entropies (complexity) will remain unchanged. If the error is a **bias**, the binned distribution of a time slice will be shifted, but its shape will remain unchanged. In this case, both uncertainty and complexity will remain unchanged.

System nonstationarity: Long term changes of the data series can either appear in the form of **trends** or **breakpoints**. Let us consider both at the example of only two time slices, each covering half of the data series as shown in Fig. R6.

[Figure]

Fig. R6

For the stationary series, entropies in both slices are equal and small, mean entropy (uncertainty) is therefore **small**, and the entropy of entropies (complexity) is **zero**.

For the trend series, entropies in both slices are equal and high, mean entropy (uncertainty) is therefore **large**, and the entropy of entropies (complexity) is **zero**.

For the breakpoint series, entropy in the first time slice is small, in the second it is large. Mean entropy (uncertainty) is therefore **medium**, and the entropy of entropies (complexity) is **large**.

Comparing the three, it seems that stationary series, trends and breakpoints in a data series will leave characteristic and distinguishable traces in a c-u-curve.

We suggest adding a sentence addressing both the influence of errors and nonstationarity to a new section 2.2 ('Properties') in a revised version of the manuscript.

**Comment 13:** What is the effect of data transformation on the inferred c-u-curve? The authors can again resort to discharge data and compare the c-u-curve of the original signal with its counterpart derived from a Box-Cox transformation. One could even consider wavelet analysis, but this is for future work.

Reply 13: If the binning is kept the same, a nonlinear transformation of the data will change the results: wherever data are squeezed, bin populations will increase, wherever data are stretched, bin populations will decrease, and this will directly affect the entropy of the binned distribution. If the binning undergoes the same transformation as the data, the results will remain unchanged. As a user is free to choose a suitable binning, including non-uniform binning, there is no real reason for data transformation, as the desired effect of stretching and squeezing data can also be achieved by the choice of the binning. We suggest adding a sentence addressing this point to a new section 2.2 ('Properties') in a revised version of the manuscript.

**Comment 14:** Equation (4): I find the choice of mathematical variables not particularly intuitive. The authors must have thought about their choice of symbols much better than I did, but why not assume at the outset that we are looking (typically) at temporal data and use Dt for the temporal extent of the time slice, n, for the number of slices and so forth. Then one can assign subscripts to these variables to differentiate between their definitions for the two metrics.

Reply 14: We agree that we should better explain our choice of variable names and will add a related sentence at the beginning of section 2.1. The reasoning behind our choice of symbols is: 'n' is for 'number, 'v' is for 'value', 'b' is for 'bin', 's' is for 'slice', 'e' is for 'entropy', 't' is for 'time step', 'w' is for

'width'. 'nvb' for instance then is 'number of value bins', which we hope is intuitive once it has been explained to the reader.

**Comment 15:** Line 97: Why should each bin of the histogram have at least some nominal number of m values? This seems rather artificial. Why not construct the histogram using the rules of thumb of Scott or Sturges? A bin cannot have zero values as this introduces difficulties with the computation of the density and log of the density in Eqs. (1) and (3)? I can only recommend googling 'How to calculate the Kullback-Leibler divergence for discrete distributions' – this will provide ways forward how to compute the product of p_i and log(p_i) if p_i is zero. Additionally, the authors can think of a Gaussian mixture model to fit a distribution through the histogram and use this fitted mixture to compute system uncertainty and system complexity. This process is sufficiently fast to warrant practical use in a long time series with many slices.

Reply 15: The potential problem mentioned by the referee, 'infinite Kullback-Leibler divergence when p(observed)≠0 and p(model)=0' does not apply here, as we calculate entropy. For entropy, there is no problem with p=0, as in such a case, the limit of 0*log(0) goes to zero. The reason we recommend assuring a certain minimum average population of each bin by ensuring a good relation between the size of the data set (nt) and the number of bins (neb) is to assure that the binned distribution is a robust representation of the underlying distribution of the data, and not hampered by limited sample size. The goal – finding a good tradeoff between resolution and representativeness - underlies all of the binning methods mentioned in reply 6, all of which can be used instead of the 'm' approach to choose an appropriate binning. In a revised version of the manuscript, we will make this clear (see also our reply 6).

**Comment 16:** Line 196: The authors refer to random noise as a purely chaotic process. I do not think this is an accurate description of random noise as draws from a normal variate do not satisfy the definition of chaos.

Reply 16: We agree. In a revised version of the manuscript, we will remove 'purely chaotic process' from the sentence.

**Comment 17:** The authors analyse the daily discharge data of two watersheds. First, I believe that hourly data is available for most watersheds. Maybe not for > 20 years uninterrupted but for sufficiently long times to satisfy the requirements of the methodology. More importantly, as a demonstration of the power and usefulness of the methodology the authors should consider using watersheds of the CAMELS data set with different hydrologic regimes (see recent classification methods published in HESS). How does the c-u-curve respond to the hydrologic regime? Do we see differences among all hydrologic regimes? And if we take multiple watersheds from the same regime, do they group in the c-u space? I consider this extension to all (4 or 5) hydrologic regimes to be important as it tests the robustness of the methodology.

Reply 17: We fully agree with the suggestions made by the referee, but for the reasons given in comment 2 we suggest addressing them in a follow-up paper. Nevertheless, we here provide some preliminary results from application of the c-u-curve to the CAMELS US dataset. Overall 666 catchments were classified into dry, wet and snow-influenced hydrological regimes by the fraction of precipitation falling as snow (fs) and the aridity index (AI=PET/P) (see Fig R7a):

- Wet Regime ($f_s<0.2$ and AI<1).
- Dry Regime ($f_s<0.2$ and AI>1).
- Snow Regime ($f_s>0.2$).

The variation of the maximum complexity for each catchment per hydrological regime is shown in R7.b. It reveals that different hydrologic regimes exhibit different variability of complexity. Moreover, the plot of complexity and entropy (R7.c) for different regimes shows the average complexity of all catchments in a hydrological regime (each dot is the average for a particular slice width). This further

highlights the differences among hydrological classes, which can possibly be attributed to differences in catchment structural properties, process inventory and hydroclimatic forcing. We will explore this in detail in the next paper.

[Figure]

Fig R7

**Comment 18:** I very much enjoyed reading this paper. From my comments above it is clear that I have concerns about the statistical/mathematical and literature underpinning of the methodology, the use of the words system uncertainty and system complexity, and the robustness and convergence properties of the methodology (= c-u curve). The additional studies I suggest will help answer important questions about the usefulness and diagnostic power of the c-u curve and its use in the analysis and classification of hydrometeorological time series. My comments are intended to help the authors further refine/improve their methodology for maximum exposure and use in the community.

PS. I did not proofread my review. Also, my comments are listed in a somewhat random order (with a c-u-curve that approaches a point) as a result of going back and forth in the paper.

Jasper Vrugt

Reply 18: We appreciate all comments by the referee! I hope we could suitably address his concerns in the above replies. With respect to robustness of the c-u-curve method, please see also our reply below on a question directly sent to us by the referee (comment 19).

**Comment 19**: (this comment was directly sent to us by the referee, and we decided to include it here): I just realize now that I forgot to comment on the choice of non-overlapping windows. What are the arguments for this - or against using overlapping windows? I am sorry for my technical questions but this should help widespread dissemination of the methodology.

Reply 19: When using overlapping windows, the data in the overlapping areas will be used twice, while the data in the non-overlapping areas will be used only once. This will give the former a higher weight in the overall results compared to the latter, for which we think there is no good justification. However, inspired by this question raised by the referee, we thought about other ways of making better use of the data to increase the robustness of the results. So far, for a particular time slicing scheme, the available data are split into time slices just once, placing each data point into exactly one slice. E.g. for slice width '3', the slices are '1-2-3', '4-5-6' etc. Depending on the particular position of the split (e.g. at the beginning or in the middle of a flood), this might lead to very high or low within-slice uncertainty, which in turn can make uncertainty and complexity subject to random fluctuation, in other words non-robust. To overcome this potential limitation of the method, we tested a moving-window approach: For each time slicing scheme, we calculated uncertainty and complexity for all possible window shifts, i.e. all shifts smaller than the width of the time slice. For the size-3 example used above, it means shift 0 = time slices '1-2-3', '4-5-6' etc; shift 1 = time slices '2-3-4', '5-6-7' etc.; shift 2 = time slices '3-4-5', '6-7-8' etc.; Shift 4 and higher are not required, because they would be shifted repetitions of previous windows. For practical reasons we assumed a circular data set, where the last value connects to the first. This avoids data loss at the beginning and end of the data set. For each window shift we calculated uncertainty and complexity, and then took the average over all shifts. The resulting c-u-curves are shown in Fig. R7. The figure shows the same c-u-curves as Fig. 2 in the manuscript, red are the original values (fixed window), black are the mean values for moving window shifts.

[Figure]

Fig. R7

In the figure, differences between the old fixed-window and the new moving-window approach are very small. This means that the fixed-window approach can be considered as robust, at least for the given data set. However, for shorter data sets, or for data sets with very high short-time variability, random effects due to fixed windows cannot be excluded.

We therefore suggest including in a revised version of the manuscript a sentence where we describe the moving window tests (but without including the above plot) and drawing the conclusion that for the given data set, the non-shifted approach is robust, but recommend a moving-window approach for shorter data (at the price of higher computational effort). In the code repository, we will provide an updated version including the moving-window option.

**Additional comment**: Pondering over Fig. R5, it also occurred to us that an upper limit of complexity as a function of uncertainty exists, which in parts (for very low and high uncertainty) is lower than the general upper limit indicated by the "max complexity" line. To explore this, we calculated c-u-pairs for all time series we had used, for many block averaging schemes, and for many slice widths, and plotted the c-u-pairs in Fig. R8. Please note that the axes in Fig. R8 are scaled to [0,1], just to give the editor and referee a flavour of a normalized version of the c-u-plot. Obviously, an upper complexity limit emerges. It arises from the fact that for very small and very high uncertainty values - which is the mean of all time slice entropies – the variability of these values – which is complexity – is limited. So an upper hull curve for complexity should arise from solving the question "what is the highest-entropy discrete distribution for which the mean is known?". This limit indeed exists, as shown by Conrad (2022), Theorem 5.12, and Example 5.13. The solution is semi-analytical, i.e. the unknown value of parameter $\beta_0$ in Eq. (5.10) is determined numerically, but the overall shape of the distribution function is analytically determined, and an exponential function of the expected values of each bin and $\beta_0$ (Eq. 5.5). In Fig. R8, this new limit is plotted as red line. Obviously, the limit serves as a useful reference, against which complexities of time series of interest can be compared. E.g. the overall area under the theoretical limit could be seen as an upper limit for total integral complexity achievable by a time series, which it would reach only if it is maximally complex for each slice width. Taking the ratio between the reference area and the area under a particular c-u-curve could then serve as a single-numbered measure of the overall complexity of a time series. We therefore suggest introducing and discussing this limit in a revised version of the manuscript in section 3.2, where we also explain the other bounds (lines 160 pp). We also suggest plotting this limit in Figs 2 and 3, and in the Appendix provide the relevant equations and procedure to calculate it.

[Figure]

Fig. R8

Yours sincerely,

Uwe Ehret and Pankaj Dey

**References**

Addor, N., Nearing, G., Prieto, C., Newman, A. J., Le Vine, N., and Clark, M. P.: A Ranking of Hydrological Signatures Based on Their Predictability in Space, Water Resources Research, 54, 8792-8812, https://doi.org/10.1029/2018WR022606, 2018.

Addor, N., Newman, A. J., Mizukami, N., and Clark, M. P.: The CAMELS data set: catchment attributes and meteorology for large-sample studies, Hydrol. Earth Syst. Sci., 21, 5293-5313, 10.5194/hess-21-5293-2017, 2017.

Conrad, K. (2022): Probability distributions and maximum entropy. Downloaded 2022/04/29 from https://kconrad.math.uconn.edu/blurbs/analysis/entropypost.pdf.

Feldman, D. P., and Crutchfield, J. P. (1998): Measures of statistical complexity: Why?, Physics Letters A, 238, 244-252, 10.1016/s0375-9601(97)00855-4.

Freedman, D., and Diaconis, P.: On the histogram as a density estimator:L2 theory, Zeitschrift für Wahrscheinlichkeitstheorie und Verwandte Gebiete, 57, 453-476, 10.1007/BF01025868, 1981.

Jehn, F. U., Bestian, K., Breuer, L., Kraft, P., and Houska, T.: Using hydrological and climatic catchment clusters to explore drivers of catchment behavior, Hydrol. Earth Syst. Sci., 24, 1081-1100, 10.5194/hess-24-1081-2020, 2020.

Knuth, K. H.: Optimal data-based binning for histograms and histogram-based probability density models, Digital Signal Processing, 95, 102581, https://doi.org/10.1016/j.dsp.2019.102581, 2019.

Kuentz, A., Arheimer, B., Hundecha, Y., and Wagener, T.: Understanding hydrologic variability across Europe through catchment classification, Hydrol. Earth Syst. Sci., 21, 2863-2879, 10.5194/hess-21-2863-2017, 2017.

Ladyman, J., Lambert, J., and Wiesner, K.: What is a complex system?, European Journal for Philosophy of Science, 3, 33-67, 10.1007/s13194-012-0056-8, 2013.

Liu, Y., and Gupta, H. V.: Uncertainty in hydrologic modeling: Toward an integrated data assimilation framework, Water Resources Research, 43, https://doi.org/10.1029/2006WR005756, 2007.

Lloyd, S.: Measures of complexity: a nonexhaustive list, IEEE Control Systems Magazine, 21, 7-8, 10.1109/MCS.2001.939938, 2001.

LopezRuiz, R., Mancini, H. L., and Calbet, X. (1995): A statistical measure of complexity, Physics Letters A, 209, 321-326, 10.1016/0375-9601(95)00867-5.

Pechlivanidis, I. G., Jackson, B., McMillan, H., and Gupta, H. V.: Robust informational entropy-based descriptors of flow in catchment hydrology, Hydrological Sciences Journal, 61, 1-18, 10.1080/02626667.2014.983516, 2016.

Scott, D. W.: On optimal and data-based histograms, Biometrika, 66, 605-610, 10.1093/biomet/66.3.605, 1979.

Sturges, H. A.: The Choice of a Class Interval, Journal of the American Statistical Association, 21, 65-66, 10.1080/01621459.1926.10502161, 1926.

Vrugt, J. A., ter Braak, C. J. F., Gupta, H. V., and Robinson, B. A.: Equifinality of formal (DREAM) and informal (GLUE) Bayesian approaches in hydrologic modeling?, Stochastic Environmental Research and Risk Assessment, 23, 1011-1026, 10.1007/s00477-008-0274-y, 2009.

Vrugt, J. A., Gupta, H. V., Bouten, W., and Sorooshian, S.: A Shuffled Complex Evolution Metropolis algorithm for optimization and uncertainty assessment of hydrologic model parameters, Water Resources Research, 39, https://doi.org/10.1029/2002WR001642, 2003.

---

## Author Comment (AC2)

**hess-2022-16 Responses to comments by referee #2**

Dear Editor, dear Referee,

We thank the second referee for the review of our manuscript. We will in the following reply to the comments point by point. The Referee comments are in blue.

**Comment 1:** The technical note is on the topic of applying information theory to the analysis of observed hydrometeorological time series (precipitation and flow). The method contributed here (as far as I understand) consists of computing the entropy over many slices (windows) of the original time series varying the width of the slices. These entropies are seen as measures of system uncertainty. A separate measure of system complexity is defined by the "entropy of entropies". The "system uncertainty" is graphed versus "system complexity" and this curve is given the term "c-u curve". This analysis is applied to several time series including from general dynamic systems (Lorenz attractor) and from hydrological systems (basin streamflow). I believe the proposed ideas hold merit and can potentially make a valuable contribution to data analysis in hydrology.

Reply 1: Thank you for the concise summary of our approach and the overall positive evaluation.

**Comment 2:** However the current presentation and its format make it difficult to ascertain the contributions. The paper is submitted in the format of a "technical note", but it gives the impression of a full paper with the introduction / literature review essentially missing, …

Reply 2: Presenting the approach as a technical note or research paper: This question was also raised by referee #1 (RC1 comment 2), so obviously this is an important point to consider. Initially we indeed thought about presenting the c-u-curve method in a full scientific paper, with the method description and a range of applications, including:

- Hydrological classification: Use data from hydrologically distinctly different catchments such as groundwater dominated, interflow dominated, dominated by reservoir operation, arid, humid and snow-influenced catchments from large data sets (e.g. Addor et al. 2017, Kuentz et al., 2017) and see if and how these differences are reflected by c-u-curve properties).
- Comparison to existing hydrological classifiers and signatures (such as those discussed in Jehn et al., 2020; Addor et al. 2018; Kuentz et al., 2017) at the example of large data sets (e.g. Addor et al., 2017). This includes evaluating the classification power of c-u-curve and the evaluation how similar or dissimilar its classifications are to those from existing classifiers.
- Use for model improvement: Test if c-u-curve characteristics can be used for targeted model improvement, either as an objective function for parameter identification during model calibration, or as a signature which may point to model structural deficiencies
- System analysis: Compare c-u-curve characteristics for input, internal states and output of hydrological systems to analyse system behaviour: Are uncertainty and complexity increasing or decreasing on the way through the system?

Looking at this list we decided it will be too much material for a single paper to introduce the method and provide all these use-cases. So instead we decided to introduce the c-u-curve method in a compact manner in a technical note, along with a few examples illustrating its properties and behaviour, and presenting the other aspects in a follow-up scientific paper. We suggest that this is the best compromise for rapid yet thorough presentation of the method. To clarify this to the reader, we suggest adding in a revised version of the manuscript a sentence to the last paragraph of section 4 ('summary and conclusions'), explaining possible further avenues of research along the above bullet-point list.

**Comment 3:** … and with important method descriptions referenced to a Zenodo archive (line 64) which is not a peer-reviewed publication.

Reply 3: Ehret (2022), the Zenodo archive to which the referee refers, is not an extra publication but a part of this manuscript. It provides all the code required to reproduce the results of the manuscript, and as such it is required by the HESS publication rules. Through the review process of this manuscript, it will become part of a peer-reviewed publication. Please note that the step from uni- to multivariate, and from deterministic to ensemble application is very straightforward (as described in 'test_c_u_curve.m' on Zenodo): It simply means to expand the input data set into the second dimension (for multivariate) and into the third dimension (for ensemble). Therefore, as mentioned in the manuscript in lines 63-65, for clarity we use the 1-d deterministic case for demonstration of the method in the manuscript, and point the reader to the demo examples on Zenodo for the other cases.

Please note that since the first submission of the manuscript, we have updated the code archive. The DOI for the new archive is https://doi.org/10.5281/zenodo.7124382. We will update the DOI in a revised version of the manuscript.

**Comment 3:** As such the entire literature review seems limited to 4-5 lines (lines 48-52). This would seem insufficient for a publication claiming to provide a general method for analysis of dynamic systems.

Reply 3: We agree that the literature review should be more comprehensive. In a revised version of the manuscript, we suggest to include and briefly discuss the work on complex system analysis in general by Ladyman et al. (2013), Lloyd (2001), LopezRuiz et al. (1995), Feldman and Crutchfield (1998), and hydrological complexity in particular by Jakeman and Hornberger (1993), Vapnik (2006), Vapnik and Chervonenkis (2015), Pande and Moayeri (2018), Omabdi et al. (2021).

Please also see our reply to comment 4, where we suggest further literature to be included in a revised version of the manuscript.

**Comment 3:** The intended scope of the contributed method is also not clearly defined. If the claimed contribution is for general systems, then the literature review should be far more general than the hydrological literature. Otherwise if the contribution is made in the context of hydrological data analysis then the title and abstract should be toned down and made more specific.

Reply 3: Like the multiscale entropy method the referee mentions in comment 4, the c-u-curve method is applicable to any dynamical system, which we point out in the abstract (lines 7-10 and 18-19) and in the summary and conclusions (lines 256-257), and which we illustrate by using three typical synthetical time series (constant, random noise, Lorenz attractor). As our domain expertise is in hydrology, and as hydrological systems - and signals coming from them - often qualify as complex (see the related explanation in lines 46 – 48), we additionally use hydrological time series to demonstrate that the related c-u-curve characteristics are in accordance with domain (here: hydrological) system understanding. We therefore suggest that we made the general scope of the proposed method sufficiently clear in the manuscript. Nevertheless, we agree with the referee that we should give a more detailed overview on dynamical system analysis in general. For this, please see our reply on comment 4.

**Comment 4:** In terms of previous work, the c-u method looks similar to the "Multiscale Entropy" methods which also look at entropies for a range of time resolutions, and also define terms of uncertainty, complexity, etc. For example a quick search indicates:

- Hu, M., Liang, H. (2017). Multiscale Entropy: Recent Advances. https://sapienlabs.org/lab-talk/understanding-multiscale-entropy/

- Wu et al (2013) Entropy, 15(3), 1069-1084; "Time Series Analysis Using Composite Multiscale Entropy; https://doi.org/10.3390/e15031069

I am not implying the methods are the same, however as a reviewer I believe if a journal paper (regardless of its format as full paper or technical note) claims as its contribution a new way of using entropy to quantify uncertainty-complexity of general dynamical systems then the onus is on the authors to conduct a thorough literature review, define the scope of the innovation, discuss advantages and disadvantages with respect to existing approaches, etc.

Reply 4: Thank you for pointing us at methods related to the concept of multiscale entropy (MSE). Indeed the c-u-curve shares with MSE the idea that the entropy values are calculated for various aggregations of the original data, and that from the joint display and comparison of these entropy values much can be learned about the underlying dynamical system. The difference is that in MSE, the aggregation is typically done by adding consecutive values, and aggregated entropies are plotted versus the time-scale of aggregation, while in the c-u-curve method, entropies are always computed from the original data, but in blocks of various sizes, and that for a given block-size, complexity is calculated as the entropy of all block-entropies. In a revised version of the manuscript, we suggest to briefly introduce multiscale entropy methods in section 1 (Introduction), and discuss their main similarities and differences with the c-u-curve method in section 2.1 (Method description), including the following references: Costa et al. (2002), Li and Zhang (2008), Guzmán-Vargas et al. (2008), Brunsell (2010), Wu et al. (2013).

Also, when revisiting the manuscript, we found another interesting property of the c-u-curve, which links to work by Conrad (2022). We suggest including a short description of the property, and the reference into a revised version of the manuscript. The following explanation of the property is copied from the reply to referee #1 (last comment).

Pondering over many versions of Fig. 3 in the manuscript, it occurred to us that an upper limit of complexity as a function of uncertainty exists, which in parts (for very low and high uncertainty) is lower than the general upper limit indicated by the "max complexity" line. To explore this, we calculated c-u-pairs for all time series we had used, for many block averaging schemes, and for many slice widths, and plotted the c-u-pairs in Fig. R1 (see below). Please note that the axes in Fig. R1 are scaled to [0,1], just to give the editor and referee a flavour of a normalized version of the c-u-plot. Obviously, an upper complexity limit emerges. It arises from the fact that for very small and very high uncertainty values - which is the mean of all time slice entropies – the variability of these values – which is complexity – is limited. So an upper hull curve for complexity should arise from solving the question "what is the highest-entropy discrete distribution for which the mean is known?". This limit indeed exists, as shown by Conrad (2022), Theorem 5.12, and Example 5.13. The solution is semi-analytical, i.e. the unknown value of parameter $\beta_0$ in Eq. (5.10) is determined numerically, but the overall shape of the distribution function is analytically determined, and an exponential function of the expected values of each bin and $\beta_0$ (Eq. 5.5). In Fig. R1, this new limit is plotted as red line. Obviously, the limit serves as a useful reference, against which complexities of time series of interest can be compared. E.g. the overall area under the theoretical limit could be seen as an upper limit for total integral complexity achievable by a time series, which it would reach only if it is maximally complex for each slice width. Taking the ratio between the reference area and the area under a particular c-u-curve could then serve as a single-numbered measure of the overall complexity of a time series. We therefore suggest introducing and discussing this limit in a revised version of the manuscript in section 3.2, where we also explain the other bounds (lines 160 pp). We also suggest plotting this limit in Figs 2 and 3, and in the Appendix provide the relevant equations and procedure to calculate it.

[Figure]

Fig. R1

**Comment 5:** The results section where the method is applied to basin precipitation and flow provides interesting ideas regarding the controls on streamflow complexity, for example lines 220-245 where the complexity is contrasted for two basins. For the benefits of the HESS audience, I would suggest more emphasis on this type of understanding would be useful.

Reply 5: We are glad the referee finds the hydrological application examples we provide in the manuscript useful. As explained in our reply to comment 2, we intend to provide an in-depth application to hydrological examples in a follow-up scientific paper. For the technical note, we believe that the mixture of general (line, random noise, Lorentz attractor) and hydrological applications (precipitation and streamflow STR, streamflow GR) is small enough to keep the technical note sufficiently lean, but suitable to demonstrate that the method is both generally applicable, and that it provides results which are in line with domain expertise when applied to data of a particular domain (here: hydrology).

**Comment 6:** Readers without a strong background in information theory would also benefit from some help on how to interpret information-theory concepts, for ex the axis scale in "bits" (eg Fig 3) and how to relate it to more common hydrological units.

Reply 6: In the description of the method in Sect. 2.1 (lines 79-82), we provide an intuitive interpretation of entropy measured in bits, but we agree with the referee that this may be too short for readers not yet familiar with the concept. In a revised version of the manuscript, we suggest adding to Sect. 2.1 a short discussion of how entropy compares to variance, a measure of spread of a

data-distribution most readers are familiar with. We also suggest adding to the captions of Fig. 2 and Fig. 3 a reference to Sect. 2.1 for more information on how to interpret the axis units.

**Comment 7:** I would also suggest helping the reader through the results and discussion sections. Perhaps identify in advance some aims for this analysis and then follow them thru. Otherwise these sections are quite monolythic and a bit hard to follow.

Reply 7: We agree and suggest adding to a revised version of the manuscript, at the beginning of section 3.2, a short overview on the structure and content of the section.

**Comment 8:** Overall I recommend a major revision to address these issues and produce a clearer and stronger contribution.

Reply 8: We hope that the changes to our manuscript as proposed in the replies to comments 1-7 meet the referee's request to make our manuscript both clearer and stronger.

Yours sincerely,

Uwe Ehret and Pankaj Dey

**References**

Addor, N., Nearing, G., Prieto, C., Newman, A. J., Le Vine, N., and Clark, M. P.: A Ranking of Hydrological Signatures Based on Their Predictability in Space, Water Resources Research, 54, 8792-8812, https://doi.org/10.1029/2018WR022606, 2018.

Addor, N., Newman, A. J., Mizukami, N., and Clark, M. P.: The CAMELS data set: catchment attributes and meteorology for large-sample studies, Hydrol. Earth Syst. Sci., 21, 5293-5313, 10.5194/hess-21-5293-2017, 2017.

Brunsell, N.A. (2010): A multiscale information theory approach to assess spatial-temporal variability of daily precipitation. J. Hydrol. 2010, 385, 165–172.

Conrad, K. (2022): Probability distributions and maximum entropy. Downloaded 2022/04/29 from https://kconrad.math.uconn.edu/blurbs/analysis/entropypost.pdf.

Costa, M.; Goldberger, A.L.; Peng, C.K. (2002): Multiscale entropy analysis of complex physiologic time series. Phys. Rev. Lett. 2002, 89, 68102.

Ehret, U.: KIT-HYD/c-u-curve: Version 1.0 (1.0.0). Zenodo. https://doi.org/10.5281/zenodo.5840045, 2022.

Feldman, D. P., and Crutchfield, J. P. (1998): Measures of statistical complexity: Why?, Physics Letters A, 238, 244-252, 10.1016/s0375-9601(97)00855-4.

Guzmán-Vargas, L., A. Ramírez-Rojas, and F. Angulo-Brown (2008), Multiscale entropy analysis of electroseismic time series, Nat. Hazards Earth Syst. Sci., 8(4), 855-860.

Jakeman, A. J. and Hornberger, G. M.: How much complexity is warranted in a rainfall-runoff model?, Water Resources Research, 29(8), 2637–2649, doi:10.1029/93WR00877, 1993

Jehn, F. U., Bestian, K., Breuer, L., Kraft, P., and Houska, T.: Using hydrological and climatic catchment clusters to explore drivers of catchment behavior, Hydrol. Earth Syst. Sci., 24, 1081-1100, 10.5194/hess-24-1081-2020, 2020.

Kuentz, A., Arheimer, B., Hundecha, Y., and Wagener, T.: Understanding hydrologic variability across Europe through catchment classification, Hydrol. Earth Syst. Sci., 21, 2863-2879, 10.5194/hess-21-2863-2017, 2017.

Ladyman, J., Lambert, J., and Wiesner, K.: What is a complex system?, European Journal for Philosophy of Science, 3, 33-67, 10.1007/s13194-012-0056-8, 2013.

Li, Z., and Y.-K. Zhang (2008), Multi-scale entropy analysis of Mississippi River flow, Stochastic Environmental Research and Risk Assessment, 22(4), 507-512.

Lloyd, S.: Measures of complexity: a nonexhaustive list, IEEE Control Systems Magazine, 21, 7-8, 10.1109/MCS.2001.939938, 2001.

LopezRuiz, R., Mancini, H. L., and Calbet, X. (1995): A statistical measure of complexity, Physics Letters A, 209, 321-326, 10.1016/0375-9601(95)00867-5.

Ombadi, M., Nguyen, P., Sorooshian, S. and Hsu, K.: Complexity of hydrologic basins: A chaotic dynamics perspective, Journal of Hydrology, 597, 126222, doi:10.1016/j.jhydrol.2021.126222, 2021.

Pande, S. and Moayeri, M.: Hydrological Interpretation of a Statistical Measure of Basin Complexity, Water Resources Research, 54(10), 7403–7416, doi:10.1029/2018WR022675, 2018.

Vapnik, V.: Estimation of Dependences Based on Empirical Data, 1st ed., Springer Science & Business Media, New York, NY. [online] Available from: https://doi.org/10.1007/0-387-34239-7, 2006.

Vapnik, V. N. and Chervonenkis, A. Ya.: On the Uniform Convergence of Relative Frequencies of Events to Their Probabilities, in Measures of Complexity: Festschrift for Alexey Chervonenkis, edited by V. Vovk, H. Papadopoulos, and A. Gammerman, pp. 11–30, Springer International Publishing, Cham., 2015.

Wu, S.-D., C.-W. Wu, S.-G. Lin, C.-C. Wang, and K.-Y. Lee (2013), Time Series Analysis Using Composite Multiscale Entropy, Entropy, 15(3), 1069-1084.

---

## Author Response (AR2)

**hess-2022-16 Responses to comments by the editor (Jim Freer) as of 28 Feb 2023**

Editor comments are in blue, our replies are in black.

Public justification (visible to the public if the article is accepted and published):

Dear Authors,

Whilst the reviewers appreciate the changes within the manuscript and that there is merit in the research presented they still note some difficulties (in their view) with the readability, a slightly dense and convoluted manuscript at times especially when introducing the core methods, and most importantly a continuing discussion about ensuring the theoretical underpinnings are clear and dealt with appropriately in the manuscript. I accept that for a technical note paper we need to ensure a strong connection to the theory and previous research that is related and I agree with the points that particularly the 1st reviewer makes. These reviewers are both considerable experts in their field. So I request again the authors consider these points raised from their revision but I do think the discussion is still at a phase where we will need to ask at least one reviewer for confirmation of how the manuscript has been changed to reflect these continuing points on the theory surrounding this research. I look forward to the authors changes to the manuscript and detailed responses to the points raised before completing one final review round, best wishes, Jim

Additional private note (visible to authors and reviewers only):

Dear Authors. I think you are improving the manuscript but we are not there quite yet. So please ensure that you respond to all the comments presented in the revision round and how you can ensure the clarity in your methods and how this relates to prior theory and developments. We do need to get that right for a technical note. I completely appreciate this can be a challenge when we have such a broad set of related research but I would still say there are some core discussion points that are being brought up that need more attention. The point here is now to either defend your position or think of how to better weave into your developments the research and nomenclature that has come before and/or is related - and thus how your methods sit and are different and why therefore they have new insights. I will be sympathetic to any strong justifications as to why you can defend your approach to your presentation but these still need to be answered. I am in agreement with your defence of this being a technical note, so I don't expect a change in the submission type, best wishes, Jim

Dear Editor,

We have updated our manuscript to improve readability according to the referees suggestions. Please see our related replies in the point-by-point replies to the referees. In terms of the questions about theoretical unterpinnings of the c-u-curve method by referee #,1, Jasper Vrugt: We have recently had the opportunity to listen a presentation of his recent work, and had the opportunity for an in-depth discussion afterwards. From that it became clear the referee focuses on questions of comparing one distribution (e.g. from a candidate model during calibration, or from a forecast) against a reference, typically observed evidence). In this context, the referee suggestions about the use, and decomposition of KL-divergence make sense. However, the goal of the c-u-curve method is on characterization of a single data set in terms of internal uncertainty and complexity, where entropy rather than KL-divergence applies. We also replied to the referee question about the connection of the law of total expectation and the c-u-curve method (comment 2): The short answer here is that we do make use of the total law of expectation, but for the (common) special case of uniform slice width the calculation of uncertainty simplifies from the equation shown in our reply to comment 2, to the E1. 2 in the manuscript. We would like to point out that overall, both of the

referees requested only minor revisions, and we hope that our related changes to the manuscript, and our point-by-point justifications where we decided to not change it, will be sufficient to make it acceptable in the current form.

Yours sincerely,

Uwe Ehret and Pankaj Dey

Dear Editor, dear Referee,

We have revised our manuscript based on the comments by the first referee, Jasper Vrugt, along the lines of our replies to the referee. In the following we will repeat the comments (in blue) together with the replies (in black). We also indicate for each comment the lines in the manuscript (in red) where we applied the changes. The line numbers are for the revised version in track change mode.

**Comment 1**: I enjoyed reading the revised manuscript. As I said in my original review, the work presented is exciting and brings up lots of new ideas that can be explored in future work. The revision satisfactorily addresses most of my original review comments, yet, I am not certain that a technical note does complete justice to the material presented. The material is interesting and important but the revision has led to a dense manuscript; I thought the original submission was easier to read. Also the construction/definition of the c-u-curve bring up other important technical (implementation) and theoretical questions that are not fully explored. This can be resolved through revision or done in future research using the present paper as reference material. I am leaning towards the second option - in part also because software is provided in a separate Zenodo DOI. The only down-side is that future research may make certain aspects of the approach incomplete/obsolete or subject to redefinition, in response to a proper theoretical foundation and redefinition of uncertainty and complexity. See my comments below.

Reply 1: We are glad about the overall positive evaluation of our revisions by the referee. We agree with the referee that this manuscript is at the edge between a technical note and a scientific paper. We have decided to publish the core method as a technical note for the reasons explained in the first round of reviews, and to present applications in a follow-up research paper. We agree with the referees preference to publish further developments of c-u-curve theory in future papers, with reference to this manuscript, because we think this manuscript presents a coherent piece of research, and to ensure timely publication. This decision comes with the need for brevity for the technical note. Mostly based on the suggestions of the referees in round one, we have revised the manuscript, which led to a substantial increase in length (e.g. the added literature overview in the introduction, the discussion of entropy vs. variance, the additional appendix with exemplary histograms, etc.). We are sure these additions have strengthened the manuscript, but at the same time we have to keep an eye on manuscript length. In terms of a potential need for a later redefinition of c-u-curve aspects, please also see our replies to comment 2.

Theoretical questions that deserve attention/further research:

**Comment 2**: 1. Now that I have accidentally educated myself a bit further on the topic of information theory as part of an article that I was working on (different topic) I would recommend the authors to look into a deeper theoretical foundation for their c-u-curve. Is it not possible to find mathematical/statistical expressions for the mean of the entropy - and the entropy of the entropy using the law of total expectation (with the entropy, H(X), and data, X, that are random variables on the same probability space). See what this decomposition brings. This is the theoretical underpinning in continuous space and leads to a discretized form.

Reply 2: The law of total expectation is used in the c-u-curve method when calculating uncertainty as the expected value of entropy (Eq. 2). In other words, what is done in Eq. 2 is that when we are asked to give a single-valued best guess of the entropy within any time slice, then the expected value of all time-slice entropies is exactly this best guess. If the time slice widths differ, then we need to consider the occurrence probability of each slice when calculating the expected value, and the width of each time slice ($n_t$) relative to the total length of the time series (T) would be an appropriate measure of

this occurrence probability. In this general case, the law of total expectation would be used to calculate uncertainty:

$$E\big(H(X)\big) = \sum_{s=1}^{ns} H_s(X|s) \cdot p(s), where \; p(s) = \frac{nt(s)}{T}$$

In the case that nt of all slices are equal (equal-width slices), this equation simplifies to Eq. 2 as shown in the manuscript. This is also mentioned in the manuscript in line 114. In this sense we would like to respond to the referee that his suggestion is already incorporated in the c-u-curve method.

**Comment 3**: 2. The authors define the mean (expectation) of the entropy as uncertainty. This definition is at odds with the common definition of uncertainty in forecasting systems - which defines uncertainty as the entropy of the mean (probabilities).

Reply 3: We termed expected entropy as "uncertainty" as it specifies, in a single number, how uncertain we are on average (over all time slices) when we have to guess a particular value within a particular time slice. Due to its formulation in information terms, this uncertainty has a very intuitive interpretation of "number of binary questions to ask, if the distribution (of values within the time slice) is known and one after another all values in the time slice have to be guessed". Multiplying the uncertainty as we calculate it in Eq. 2 with the total number of points nt of the time series yields exactly the total number of questions needed to be asked to guess all values in the time series, if for each value we know the distribution of the time slice it is in. Interpreting entropy as a measure of uncertainty is at the very heart of information theory and was not invented or coined by us, and it is to our knowledge not add odds with the use of uncertainty in forecasting systems, where the uncertainty of a particular forecast can be measured by the spread of the prediction ensemble via the entropy of the ensemble.

**Comment 4:** 3. I recommend the authors to have a look at the decomposition in Weijs et al. (2010) of the KL-divergence. This is based on earlier work of Murphy (1973) on decomposition of the Brier Score. This results in 3 terms; uncertainty, reliability and resolution.

Reply 4: After the referees presentation of his recent work in the "paper club infotheory", we now understand much better what he means by this comment. One focus of the referees work is on evaluating probabilistic forecasts against evidence (typically crisp, but could also be probabilistic), and for this purpose KL-divergence is an appropriate measure, and it can be decomposed as indicated by the referee. However the purpose of the cu-curve method is not the comparison of one data set versus another, but characterization of a single data set in terms of internal uncertainty and complexity. We therefore agree with the referee that it is worth looking at the decomposition of KL-divergence, but that this does not specifically apply to the cu-curve method.

**Comment 5:** 4. The theorem/proof the authors provide on P.14 is mathematically/statistically delinquent. First, the authors talk about a probability distribution "p" on S. This should be a probability measure (sums to 1). This measure should be part of a convex class P and be quasi-integrable with respect to all P? Formula B2 warrants a reference.

Reply 5: As indicated at the beginning of Appendix B, we here as closely as possible repeat Theorem 5.12 from Conrad (2022), and therefore also adopted the notation of Conrad (2022), where $p$ is a discrete probability distribution on $p_j$ possible states. We suggest keeping to this notation for easier linkage to the rest of Conrad (2022). Also, as we make clear that all of Appendix B2 just repeats Conrad (2022), we suggest that it is clear for the reader that formula B2 is also from that source.

**Comment 6**: 5. In the comparison to existing methods the authors bring up the KL-divergence. This is equal to the divergence of the logarithmic score, which in turn has as its generalized entropy function

negative Shannon entropy. This is the theoretical link between what is presented in the paper and Lopez-Ruiz.

Reply 6: Using KL-divergence as a measure of disequilibrium between the system and a maximum entropy benchmark was introduced Feldman and Crutchfield (1998), replacing the sum of squared differences used by Lopez-Ruiz (1995). Also, we would like to point out that there are fundamental differences between the complexity measures proposed by Lopez-Ruiz (1995), Feldman and Crutchfield (1998) and our approach, see lines 250-255 in the manuscript: "… but the essential differences of CLMC and the c-u-curve methods remain: Firstly, the former defines complexity as the product of two separate system characteristics, of which one is the departure from a benchmark system, the latter derives both characteristics from the system alone. Secondly, the former does not take the order of the data into account, while the latter explicitly does when calculating entropy for data within temporally neighbouring data within time slices."

**Comment 7:** 6. If the authors generate ensembles with the Lorenz attractor by assuming some distribution for the parameters. This would yield a distribution forecast from which the entropy can be computed. This provides an entropy at each time; how does this relate to the averaged entropy of the time slices? And how would the c-u-curve look?

Reply 7: We agree that it would be interesting to apply the cu-curve method to probabilistic time series, e.g. coming from the Lorenz attractor applied with a distribution of parameters, or from an ensemble weather forecasting system. This is indeed possible, and we provide examples thereof in the example applications in Ehret (2022), but for brevity do not present such an example in the paper (we mention this now in more detail in lines 223-229). We will do so in the follow-up research paper we are currently working on. Indeed we think it is one of the strengths of the method that the extension to multivariate and probabilistic cases is seamless: When moving from univariate to multivariate cases, the entropy within a time slice simply changes from uni- to multivariate entropy. When moving from deterministic to probabilistic variables, for each time step in a time slice, a value distribution rather than a crisp value will be used to populate the distribution of all values in the time slice, but the result will still be a single distribution with a single entropy value, which can be plotted as before in the c-u-curve. If the spread of the ensembles at each point in time is large compared to the spread of values over time within the time-slice, the overall time-slice entropy will be dominated by the ensemble spreads. In short, uncertainty of the c-u-curve method measures uncertainty in time as well as uncertainty due to ensemble spread at one point in time.

Technical/presentation questions

**Comment 8:** 1. I do not find the use of symbols intuitive. Each time when I read "nbv" I think this involves multiplication of n, b and v. Also, strictly speaking if variables are acronyms themselves they should be written in regular script? I leave this to the authors, but personally would prefer just picking regular symbols.

Reply 8: We agree with the referee that "nvb" can be mistakenly interpreted as a product of three variables, and that this misinterpretation should be avoided. We do so in the manuscript by using the "·" symbol each time a multiplication is done. For clarification, we have added a related explanation at the beginning of section 2.1 (lines 90-93). We also considered using subscripts for all but the first symbol (e.g. "$n_{vb}$" instead of "nvb") to avoid confusion with multiplication, but this partly leads to double-subscripts which are hard to interpret as well (e.g. $x_{vb}$ in Eq. 1 would become $x_{v_b}$). Also, we consulted fellow mathematicians on the appropriateness of our use of symbols for variables. Their feedback was that the use of symbol combinations for variables (such as "nvb") is not standard, but fully acceptable as it is an unambiguous notation, and because the use of the symbols is explained at the beginning of section 2.1, and each variable is explained in the text directly where it was introduced in an equation (e.g. "nvb" is explained below Eq. 1). We therefore prefer to keep the use of symbols as is.

**Comment 9:** 2. A technical note has a limited number of words. As a result, the authors have to be short in their description of the case studies. Personally, I think the paper would be easier to read if those details were presented. For instance, the Lorenz attractor. The authors refer to a code (Line 256), but further details are missing. As a result, on Line 257-258 the write that "From its three variates … only the first one is shown…". Not all readers may be familiar with the Lorenz 1963 model - nor with the computational implementation, which, by itself is not trivial (different codes do not always produce the same chaotic behavior!!); certainly they may not know the three variates, etc. Of course the software will help. But it illustrates my struggle of wanting to accept this paper with minor revision, but at the same time wanting to make sure there is enough detail for readers to understand what is presented.

Reply 9: For the general topic of "technical note" or "scientific paper", and the need for brevity, please see our response to comment 1. About the Lorenz attractor in particular: While a more in-depth explanation of the Lorenz attractor would be interesting, we argue that it is not indispensable for this manuscript, because we only use the time series as a nice example of a complex time series. The complexity of the series is directly visible from Fig. 1(c), and no deeper understanding of the Lorenz attractor is required for this. For the interested reader, we provide both the reference to the original paper (Lorenz, 1963) and the code and settings we used to generate the data (Moiseev, 2022), which we think balances the need to inform the reader with the need for brevity.

**Comment 10:** 3. Is unit "bit" or "bits"?

Reply 10: Thanks for raising this point, we realized we were inconsistent with the use of "bit" and "bits". The unit is "bit", and whenever a reference is made to the unit in general, the singular is used (e.g. "entropy is measured in bit"). Whenever a reference is made to the entropy of a particular quantity, the plural is used (e.g. discharge has an entropy of 5.3 bits"). We have checked the manuscript once more for consistent use of "bit" or "bits", and updated it where necessary.

**Comment 11:** 4. Line 462: replace computer with iterative algorithmic recipe?

Reply 11: Thanks. We modified the sentence to "The value of $\beta$ can be numerically approximated with an iterative algorithmic recipe and Eqs. B1 and B2 (see example 5.14 in Conrad, 2022)." See lines 478-479.

**Comment 12:** 5. Line 455: with given \overline{E} and maximum entropy?

Reply 12: Agreed. We removed "having" from the sentence (line 472).

**Comment 13:** 6. Equation (1): "X" -> is not defined; the sample data within the bin/slice. Or "X" is the random variable of interest of which small x are the samples? Also should log_2 not be upright as it is a mathematical function?

Reply 13: Thanks. X is indeed the entire sample data in the slice, and x are the subset of X falling into a particular value bin. We have added a definition of X in line 101.

Log_2: Thanks, we changed it to upright, also in Eq. (3)

**Comment 14**: 7. Line 98: If slices are not of uniform width - would this not create difficulties with entropy comparison/averaging, etc.?

Reply 14: Yes, in such a case calculating uncertainty according to Eq. 2 should consider the relative contribution of each slice (by weighting its entropy with the slice width relative to the total length of the time series). Surely in most cases uniform slice widths are preferable, the point we wanted to make with our statement is that choosing non-uniform slice widths is not impossible per se.

**Comment 15**: In summary, I enjoyed reading this revision - I have stated my concerns (that is why I rate the manuscript as good instead of excellent), but believe that the ideas presented are worthy of a prompt publication.

Reply 15: Again we thank the referee for the overall positive evaluation of our manuscript, and appreciate his comments and suggestions that led to a substantial improvement of the manuscript.

**hess-2022-16 Responses to comments by referee #2 (report #2 as of 18 Jan 2023)**

Dear Editor, dear Referee,

We have revised our manuscript based on the comments by the second referee along the lines of our replies to the referee. In the following we will repeat the comments (in blue) together with the replies (in black). We also indicate for each comment the lines in the manuscript (in red) where we applied the changes. The line numbers are for the revised version in track change mode.

The revised paper has addressed the major concerns raised in previous round of reviews. I have a few relatively minor remaining concerns:

**Comment 1:** 1. In the introduction, around lines 35-40 it is stated that "Interestingly, despite its importance and widespread use there is to date no single agreed-upon definition and interpretation of complexity". The newly added references there are helpful, but what is missing is at least some brief summary of what these references actually say. Such summary would help support the earlier statement.

Reply 1: Agreed. We have added brief summaries of Gell-Mann (1995), Lloyd (2001), Prokopenko et al. (2009) and Ladyman et al. (2013) to the introduction in lines 39-46.

**Comment 2:** 2. The presentation around lines 85-90 is rather convoluted and confusing. For example:

- it starts by citing the Xenodo repository (which is essentially supposed to serve as a repository and appendix to the paper) where the method is generalized before actually being presented in the paper itself.

- there are various sentences such as "Also, we calculate discrete entropy based on a uniform binning approach" the significance of which is unclear based on the presentation thus far.

- "Nevertheless, the method can also be used with non-uniform binning or continuous representations of data-distributions" - I am not sure how acceptable is this claim without actual proof/demonstration, especially once again before the method itself is even presented.

I would recommend that early part of the presentation be revised for clarity, e.g. that statements / generalizations that only make sense after the presentation is complete be moved to the Discussion section, speculative statements be removed or identified as such, etc.

Reply 2: Agreed. We moved most of this introductory section to the discussion of properties in section 2.2 (lines 223-238), such that the method is first introduced, and then the generalizations and limitations are discussed. We also added some more explanations to the text.